# Predicting cell type-specific epigenomic profiles accounting for distal genetic effects

Alan E. Murphy [1,2] ✉, William Beardall[3], Marek Rei[4], Mike Phuycharoen[5] & Nathan G. Skene [1,2] ✉

Understanding how genetic variants affect the epigenome is key to interpreting GWAS, yet profiling these effects across the non-coding genome remains challenging due to experimental scalability. This necessitates accurate computational models. Existing machine learning approaches, while progressively improving, are confined to the cell types they were trained on, limiting their applicability. Here, we introduce Enformer Celltyping, a deep learning model which incorporates distal effects of DNA interactions, up to 100,000 base-pairs away, to predict epigenetic signals in previously unseen cell types. Using DNA and chromatin accessibility data for epigenetic imputation, Enformer Celltyping outperforms current best-in-class approaches and generalises across cell types and biological regions. Moreover, we propose a framework for evaluating models on genetic variant effect prediction using regulatory quantitative trait loci mapping studies, highlighting current limitations in genomic deep learning models. Despite this, Enformer Celltyping can also be used to study cell type-specific genetic enrichment of complex traits.

Recent large-scale genetic efforts, genome-wide association studies (GWAS), have identified associated variants for a myriad of diseases[1–3]. However, these often lie in non-coding, regulatory regions and cannot be associated with any functional outcomes. For example, recent GWAS have found more than 90% of Single Nucleotide Polymorphisms (SNPs) reside in non-coding regions[4]. One of the main challenges to understanding the function of these regulatory variants is that gene regulatory mechanisms are highly cell type-specific[5]. Genetic sequence variants which are associated with the function or activity of regulatory elements, quantitative trait loci (QTLs), will tend to exert their effects in a cell type-specific manner. Thus, understanding the cell type-specific effect of genetic variants in epigenetic regulation would help identify the biological processes they act on.

Mapping out molecular and regulatory QTLs comprehensively in disease-relevant cell types would enable the interpretation of functional outcomes of genetic variants on gene expression and regulation.

Techniques to study QTLs include CRISPR interference (CRISPRi)[6] and Massively Parallel Reporter Assays (MPRAs)[7]. However, these approaches lack the in vivo capabilities or scalability to large sample sizes thus limiting their efficacy. Currently, the most suitable approach is to conduct population studies measuring individuals' genomic variance and its correlation to regulatory elements, i.e. xQTL mapping studies. However, epigenetic QTL studies have proven less popular than that of gene expression (eQTL mapping studies), which include full catalogues of studies with sample sizes in the high hundreds[8] and some reaching as many as 30,000[9]. The largest epigenetic QTL study was conducted in blood immune cell types[10], with a sample size of just 196, far less than the average expression QTL study. Moreover, the study is one of just a small number of histone mark (hQTL) or transcription factor (tfQTL) studies conducted. Robustly identifying regulatory QTLs from primary data in epigenetic QTL studies requires very large sample numbers due to the high-dimensional number of pairwise tests for associations between all

[1]UK Dementia Research Institute at Imperial College London, London W12 0BZ, UK. [2]Department of Brain Sciences, Imperial College London, London W12 0BZ, UK. [3]Department of Bioengineering, Imperial College London, London SW7 2AZ, UK. [4]Department of Computing, Imperial College London, London SW7 2RH, UK. [5]Division of Informatics, Imaging & Data Sciences, University of Manchester, Manchester M13 9PL, UK. ✉e-mail: a.murphy@imperial.ac.uk; n.skene@imperial.ac.uk

SNPs and all regulatory elements. Thus, undertaking such studies in large sample sizes and on every possible cell type of interest is impractical.

An alternative is to predict the effect of genetic variants using machine learning approaches, known as in silico mutagenesis. Numerous machine learning approaches have been developed to predict the effect of genetic variants across different tasks such as transcription factor binding[11,12], histone marks[13] and gene expression[14] profiles. The main advancement of more recent approaches in the field has focused on the amount of DNA sequence which models can take into account simultaneously to make the prediction. This has generally led to greater performance on held-out genomic positions and better genetic variant effect predictions. Intuitively, these larger window sizes mean that when predicting the profile at a given location, distal effects of DNA sequence can be modelled. The maximum distance from the predicted profile to DNA sequence in either direction of such models has increased from 500 base-pairs for DeepSea[15] to 20,000 base-pairs in Basenji2[13] and in the current best-in-class approach, Enformer[16], to 100,000 base-pairs. Enformer swapped out convolutional layers from Basenji2's original architecture with multi-headed attention layers and three custom, relative positional encoding functions to dramatically increase the receptive field.

Importantly, all of the aforementioned models were trained solely on DNA sequence and thus cannot predict cell type profiles that were not seen in the training phase. It would be beneficial to predict epigenetic profiles and the effect of genetic variants on these in previously unseen cell types where collecting cell type epigenetic profiles is invasive, time-consuming and costly. The necessity of such cell type epigenetic imputation is apparent even in large consortia like ENCODE's[17] data repositories where, despite hosting greater than 10,000 assays, cell types have not been experimentally assayed for every epigenetic mark of interest. Moreover, the cell types present, mostly whole tissues or cell lines, do not capture the breadth of cell types studied in disease research, for example in the brain. Some past models have attempted to address this by generalising predictions of epigenetic profiles to other cell types using chromatin accessibility information from the cell type of interest, often DNase-Seq or ATAC-Seq, such as Leopard[12], Catchitt[11] (both predict transcription factor binding profiles) and Epitome[18] (histone mark profiles). Notably, these approaches do not consider distal genetic information, instead using local genetic windows, far shorter than that of Enformer, and other local features to the infer the cell type variability. However, this local information may not be enough for a model to fully capture cell type representations.

Here, we propose Enformer Celltyping, a self-attention based neural network model to predict histone mark activity in previously unseen cell types. Our model predicts six histone mark profiles from DNA sequence, using chromatin accessibility information from the cell type of interest and is trained on 104 ENCODE samples, hereafter referred to as cell types despite containing tissue samples, isolated cell types and cell lines, sourced from EpiMap[19]. We use a custom transfer learning approach on a pre-trained Enformer[16] model to account for the distal effect of genetic code on these regulatory signals. Enformer Celltyping also embeds, using a method similar to that of Avocado's[20] and taking inspiration from the field of NLP[21], a genome-wide and local representation of cell types from chromatin accessibility information. This work makes the model applicable to any cell type of interest where chromatin accessibility data is available. We validate Enformer Celltyping's performance against best-in-class approaches and in a number of differing biological settings and inspect the cell type-specific information the model learns. Moreover, we apply a new framework to validate the predictive performance of epigenetic deep learning models at genetic variant effect predictions using QTL studies with known associations and accounting for linkage disequilibrium (LD)[22], creating a benchmark for use by the broader field. Researchers hoping to study the epigenome and the effect of genetic variants on

cell or sub-cell types not captured in repositories like ENCODE, can not use current models or have to use the closest available cell type from current model's outputs as a proxy. Here, we take a step toward addressing this problem by creating Enformer Celltyping, a model which can predict epigenetic signals in previously unseen cell types, using a large receptive field to capture genomic regulatory information. As well as the developed framework to benchmark methods for genetic variant effect predictions, we demonstrate how our model can be used to study cell type-specific genetic enrichment of complex traits. We have made a pre-trained version of Enformer Celltyping available along with a fine-tuneable version of the Enformer model, something which was previously not available, so others may make use of these in their research (code: https://github.com/neurogenomics/EnformerCelltyping[23] and pre-trained model: https://figshare.com/projects/Enformer_Celltyping/159143).

## Results

### Accurate epigenetic predictions in unseen cell types

Enformer Celltyping uses Enformer[16] through a transfer learning approach to expand on its capabilities, predicting in previously unseen cell types (that were not part of the training set). Enformer Celltyping benefits from Enformer's attention layers which produce its large receptive field (approximately 100k base-pairs), accounting for distal DNA regulatory elements in its predictions (Fig. 1a and Supplementary Fig. 1). We measure the sensitivity of this receptive field experimentally with simulated DNA sequences (Fig. 1b). The need for such large receptive field of distal regulatory elements can be inferred by observing the distance between SNPs and the peaks they regulate in histone-QTL studies (Supplementary Fig. 2): while these studies are confounded by LD, the prevalence of long-range interactions does not appear to be consistent with a model whereby most histone QTLs act via local motif disruption. Enformer Celltyping can predict in new cell types, imputing their epigenetic signal, by embedding global and local chromatin accessibility (ATAC-Seq) signals for the cell type of interest. We experimentally show how the model is also sensitive to permutations of the local chromatin accessibility with simulated signals, following a similar trend to the DNA sequence (Fig. 1c). Using this combined approach, Enformer Celltyping shows best-in-class performance at predicting six histone mark signals for any genomic region of interest (Fig. 1d).

Enformer Celltyping, when predicting the histone mark signals, embeds a global as well as the local representation of the inputted chromatin accessibility signal for the cell type of interest. This global cell type embedding shows co-localisation of similar cell types (Fig. 2a). Using this combination of global and local chromatin accessibility signal improved the model's predictive performance (Supplementary Fig. 3), though we note that the local chromatin signal had a larger effect on model performance than the global. Enformer Celltyping is capable of making accurate predictions in these previously unseen cell types using chromatin accessibility data and DNA, such as that for microglia at the transcriptional start site of the Allograft inflammatory factor 1 (*AIF-1*) marker gene where a microglia-specific H3K27ac peak was distinct from the predictions for the other brain cell types and replicated in the experimental data (Fig. 2b). Enformer Celltyping's cell type-specific predictive capabilities was notably aided by a pre-training step (Supplementary Fig. 4). This step included training the DNA and chromatin accessibility inputs as separate submodules; the DNA module predicting the average histone mark signal and distribution of signal across the training set by incorporating Enformer's pre-trained layers and the chromatin accessibility module predicting the difference of the average and cell type of interest. The pre-training step sensibly initialised the weights of the chromatin accessibility layers before combining with the DNA layers which contained the Enformer architecture for the full training step (see the Methods section Enformer Celltyping architecture).

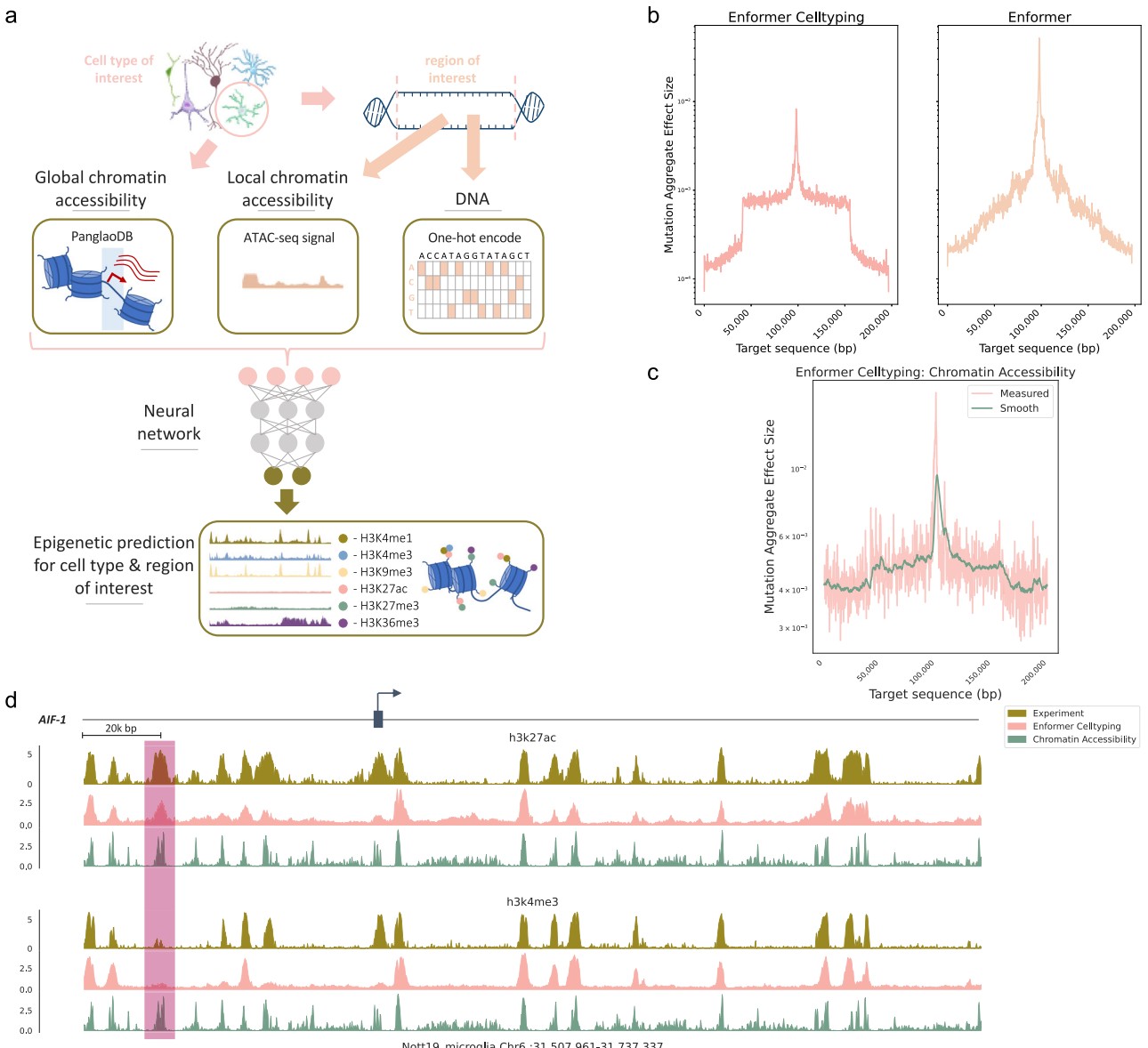

**Fig. 1 | Enformer Celltyping predicts cell type-specific histone marks accounting for distal DNA regulation. a** Enformer Celltyping uses transformer modules on DNA sequence and embeds both local and genome-wide chromatin accessibility signal to predict histone mark signals in cell types outside of the training set. Image created with the help of SciDraw (https://scidraw.io/)[74–77] under creative commons license (CC-BY). **b** The experimentally derived receptive field for both Enformer and Enformer Celltyping. Random permutations are made at increasing distances from the centre point (100,000 on the x-axis) and the effective change in the prediction on that central point is measured (y-axis). **c** Experimentally derived local chromatin accessibility receptive field for Enformer Celltyping following a similar permutation approach to (**b**). **d** An example of the observed (Experiment), predicted (Enformer Celltyping) and the chromatin accessibility signal at the transcriptional start site (blue arrow) of the Allograft inflammatory factor 1 (*AIF-1*) marker gene in microglia. The y-axis denotes the -log₁₀ *p*-value, indicating the statistical significance of a protein binding at a genomic position. Highlighted in red is a H3K27ac-specific peak not found in H3K4me3 highlighting that the model uses more than chromatin accessibility signal to identify peaks. Source data are provided as a Source Data file.

## Enformer Celltyping outperforms best-in-class approaches

We benchmarked Enformer Celltyping against Epitome[18], the current best-in-class approach for predicting histone mark signals in previously unseen cell types, in a genome-wide prediction task on three immune cell types (Fig. 3a). Enformer Celltyping (and Epitome) generalise predictions of epigenetic profiles in cell types which were not seen during training by using their chromatin accessibility information as input. To avoid the performance of our predictions being inflated for this and subsequent benchmarks, we tested three immune cell types (Monocytes, Neutrophils and T-cells) for which data was generated by ENCODE, but which we held out of the training dataset for Enformer Celltyping. For a fair comparison, we binarised the prediction of our model to match that of Epitome (see Methods section

Benchmarking). We report the ROC curves and the average precision (AP) score for both models. We evaluated performance based on the AP for each cell type and histone mark combination to get a balanced measure of the models' performances. Enformer Celltyping out-performed Epitome across all cell types and histone mark predictions.

Due to differences in predictive resolution between the two models, large genomic intervals (3,200 base-pairs) had to be used for the classification task so performance could be compared. This would result in suboptimal performance for both models, blurring their signals. To get a clearer picture of how well Enformer Celltyping predicts, following the approach of previous work[5,24], we benchmarked Enformer Celltyping's performance against the average signal (Avg) for the

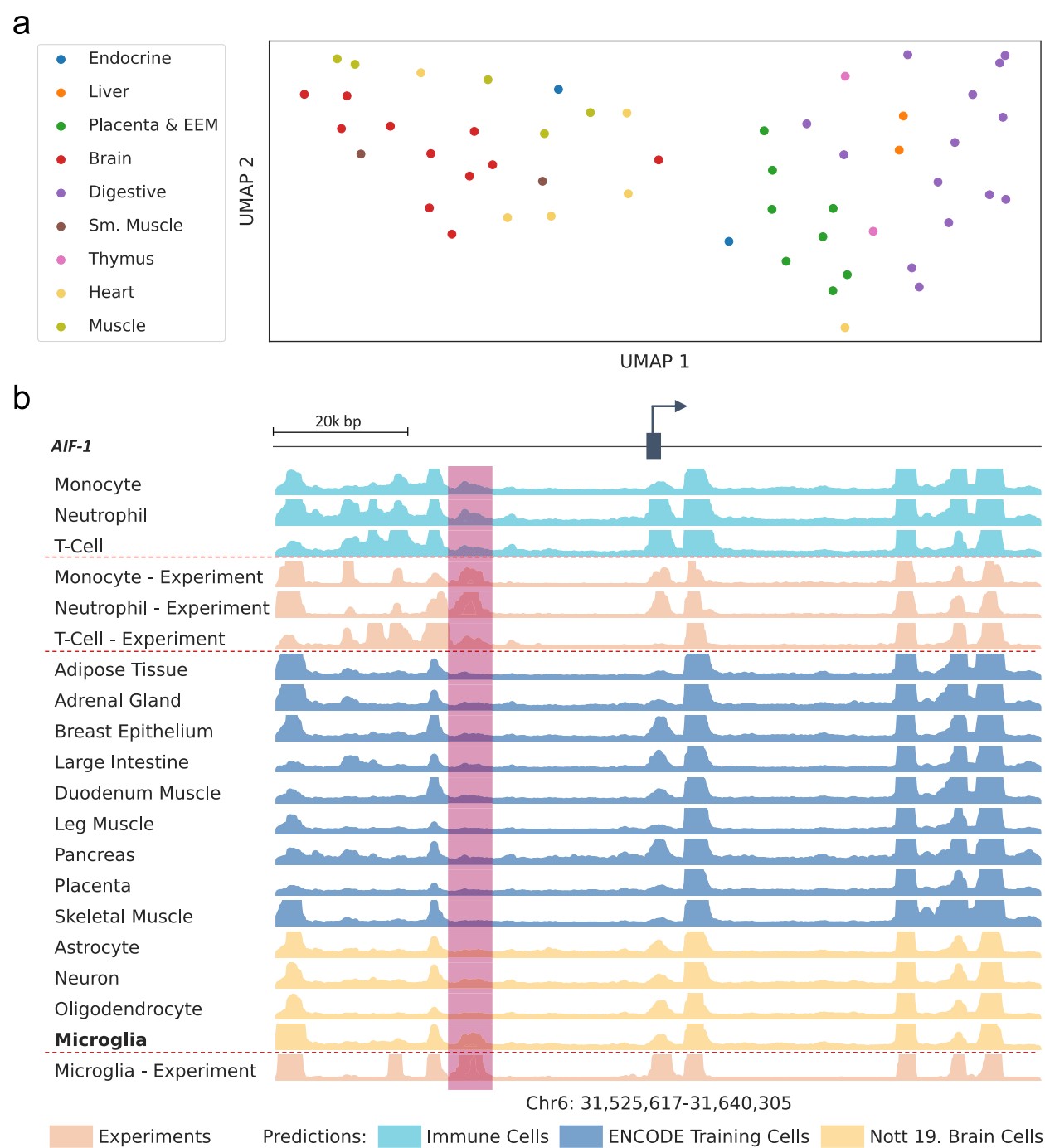

**Fig. 2 | Enformer Celltyping predicts cell type-specific histone marks signals.**
**a** A UMAP projection of Enformer Celltyping's global chromatin accessibility embeddings. Cell types from EpiMap are coloured by their broad tissue type.
**b** Enformer Celltyping's H3K27ac predictions (turquoise, navy and gold) and the experimental assay signals (apricot) at the transcriptional start site of *AIF-1* gene across varying cell types. *AIF-1* is the canonical marker of microglia. Highlighted in red is a microglia-specific peak not found in the other brain cell types but present in immune cells and which is validated in the experimental results. Source data are provided as a Source Data file.

same histone mark from the training dataset used to train Enformer Celltyping (Fig. 3b). Here we used the same three immune cell types but benchmarked performance based on the mean squared error (MSE) between the prediction and the actual -$\log_{10}$ *p*-value score, genome-wide at 128 base-pair resolution, utilising Enformer Celltyping's regression output. We split performance by each histone mark, avoiding reporting a single performance score, to investigate if the model had any specific biases. Overall, Enformer Celltyping performed similarly to the average signal. The predictions for H3K27ac and H3K4me3 signals across the three cell types does consistently out-

perform the average, with lower MSE, whereas predictive performance on H3K27me3 consistently under-performs.

One explanation for the inconsistent out-performance relative to the average, is that a model can generally perform well genome-wide by predicting the average signal. To explore this further, we note that during pre-training one of the Enformer Celltyping submodules is explicitly trained to predict the average: it's performance is shown in Fig. 3 as "EC Avg". This pre-training step was necessary as Enformer Celltyping uses Enformer as a pre-trained model for the DNA input module. The Enformer-derived layers had sensible weight

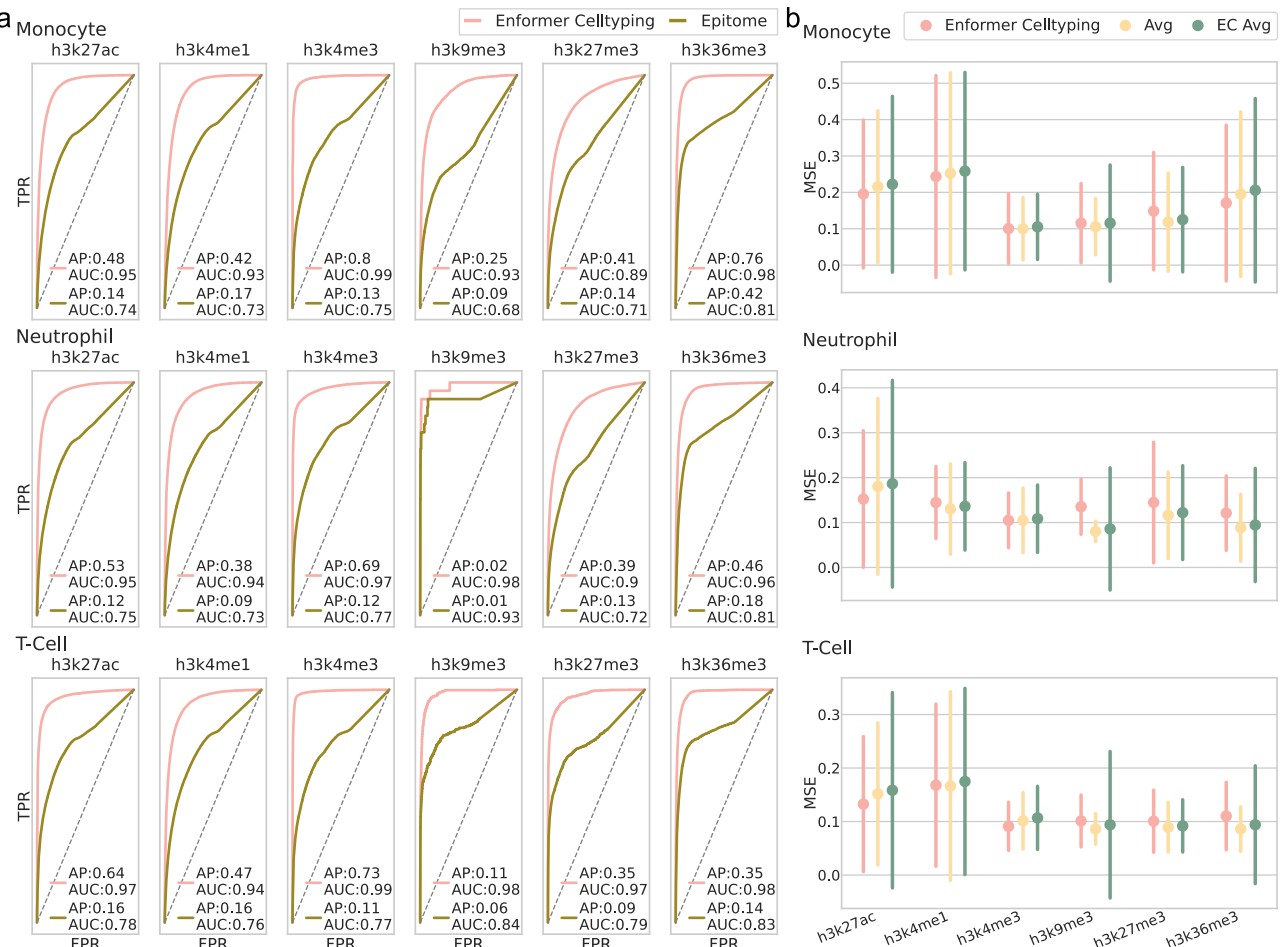

**Fig. 3 | Enformer Celltyping makes best-in-class cell type-specific histone mark predictions. a** Enformer Celltyping's outperforms Epitome at genome-wide epigenetic predictions for all three immune cell types (held out from model training). Enformer Celltyping's predictions were binarized and averaged to 3200 base-pairs to enable a comparison with Epitome. The receiver operating characteristics (ROC) curves with the area under ROC score and the average precision (AP) score are reported. **b** Enformer Celltyping frequently, but not consistently, predicts cell type-specific values (for three immune cell types) better than the average signal of the training cell types. The three cell types were not in the training data used to calculate the average. Moreover, Enformer Celltyping's superior performance against the average predictions from pre-training (EC Avg) highlight the model's ability to pick up on cell type-specific signals. Performance is measured by the mean squared error (MSE) for genome-wide predictions with error bars for the standard deviation of the error of the predictions genome-wide ($n = 8086$ genomic loci for each histone mark, cell type and model combination). Source data are provided as a Source Data file.

initialisations whereas the model's chromatin accessibility module weights were randomly initialised. Thus the short pre-training step avoids issues where all predictions were made based on the DNA information and acts as a warm-up step for the chromatin accessibility layers. During pre-training, the DNA input module predicted the average and distribution of histone mark signals across the training cell types and the chromatin accessibility input module predicted the difference between the average signal and the cell type specific signal (see Methods section Enformer Celltyping architecture and Supplementary Fig. 1a). Subsequent layers combining the two modules were next added for the second, full training step (Supplementary Fig. 1b). This pre-training step improved the overall performance of Enformer Celltyping (Supplementary Fig. 4), as did including the distribution of histone mark signals as well as the average in the DNA module predictions (Supplementary Fig. 5).

### Strong performance in functionally relevant genomic regions

While these results show Enformer Celltyping performs similar to the average signal, this is genome-wide performance, something that the average signal would be expected to perform strongly at given the overlap between cell's histone mark signals. Moreover, in Fig. 4a, we highlight that these MSE values, for both the average

signal and our model, are lower in regions of no genomic activity ('No peak' regions) which contribute to the genome-wide scores. Thus, to investigate Enformer Celltyping's performance in functionally relevant regions, we measured the model's correlation, which avoids issues of data scale and focuses on the shape i.e. is there a peak present, with the true signal at promoters, local and distal regulatory regions (Fig. 4b, c). Here, we focused on the histone marks with known functional roles in these regions; H3K27ac at active enhancers and promoters, H3K4me3 at active promoters and H3K4me1 at active enhancers[25], and derived the biological regions from Search Candidate cis-Regulatory Elements by ENCODE (SCREEN)[26], available from: https://screen.encodeproject.org/. This work highlights Enformer Celltyping's superior performance in functionally relevant regions, where the average signal fails to capture the cell type-specific signals. This is particularly notable for distal enhancer regions for the blood immune cell types where Enformer Celltyping outperformed the average in 96% of the cell type-chromosome comparisons (Fig. 4c, panel 3). Distal regulatory regions are highly cell type-specific thus highlighting the model's utility for epigenetic imputation. This distinction is further emphasised by the genome plots of selected SCREEN distal enhancer regions for the immune cell types (Supplementary Fig. 6).

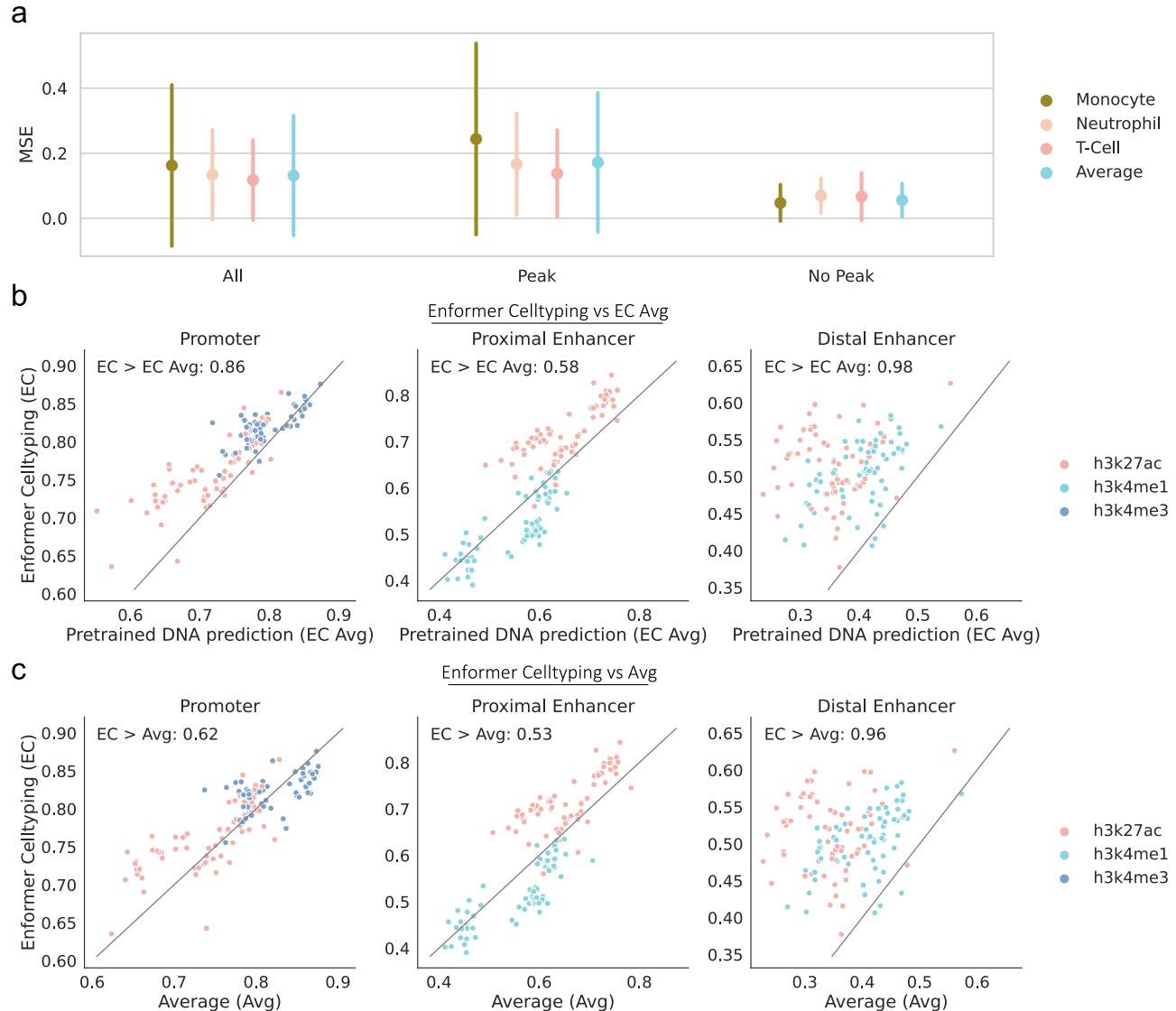

**Fig. 4 | Enformer Celltyping has superior performance at functionally relevant regions. a** Enformer Celltyping's (EC) mean squared error (MSE) genome-wide ('All') and regions of histone mark activity ('Peak') or no activity ('No Peak'). Peak and No Peak regions were derived based on a -$\log_{10}$ $p$-value cut-off of $2^{19}$. Performance is shown against the average signal of the training cell types, for the three immune cell types excluded from the training dataset. The bars denote the standard deviation of the error of the predictions across all predicted regions ($n = 8086$ genomic loci for each histone mark, cell type and model combination) that were labelled with the given activity. **b, c** Enformer Celltyping's Pearson R correlation with the true signal (y-axis) compared against the average predictions from pre-training's (EC Avg) **(b)** or the training set experimental values average's (avg) **(c)** Pearson R (y-axis) for the three blood cell types across different functional regions sourced from Search Candidate cis-Regulatory Elements by ENCODE (SCREEN)[26] and derived from a diverse range of ENCODE cell types. Performance was measured for histone marks with known functional roles in these regions for the same three immune cell types. Pearson R values were calculated per chromosome for each cell type (representing a dot in the figure). The 'EC > EC Avg' or 'EC > Avg' represents for how many cell type-chromosome values our model outperformed the alternative in. Source data are provided as a Source Data file.

Moreover, from Fig. 3b, we can also see Enformer Celltyping's cell type-specific predictions (Enformer Celltyping) outperform that of the average predictions from pre-training (EC Avg) in the vast majority of cell types and histone marks. Moreover the variance in the error of the predictions genome-wide is far smaller. However, the difference is minimal and based on genome-wide measure of MSE, which, in the same manner as for the training set average, EC Avg would be expected to perform strongly in. Thus, we also compared Enformer Celltyping's performance in functionally relevant regions against EC Avg (Fig. 4b). Enformer Celltyping outperformed EC Avg in all categories, with higher correlation with experimental signal in 98% of the comparisons tested for distal enhancer regions (Fig. 4b, panel 3), highlighting how the model picks up on cell type-specific signals during the full training step.

## Consistent performance across diverse cell types

Up to this point, Enformer Celltyping's performance has only been tested on cell types generated from ENCODE. We next measured its performance in a manner that replicates how researchers can use Enformer Celltyping to predict histone marks in their cell types of interest for which they have chromatin accessibility information[24]. To this end, we utilised H3K27ac and H3K4me3 ChIP-Seq and ATAC-Seq data for isolated cell types (PU.1+ microglia, NeuN+ neuronal, OLIG2+ oligodendrocyte and NeuN- LHX2+ astrocytes) from resected cortical brain tissue[27]. Consensus peak $p$-value tracks (taking individuals as replicates) were derived and the data processed in the same manner as for the training cell types. Enformer Celltyping has not been trained on any isolated cell types from the brain and this data was not collected following the ENCODE protocols so its performance across the two

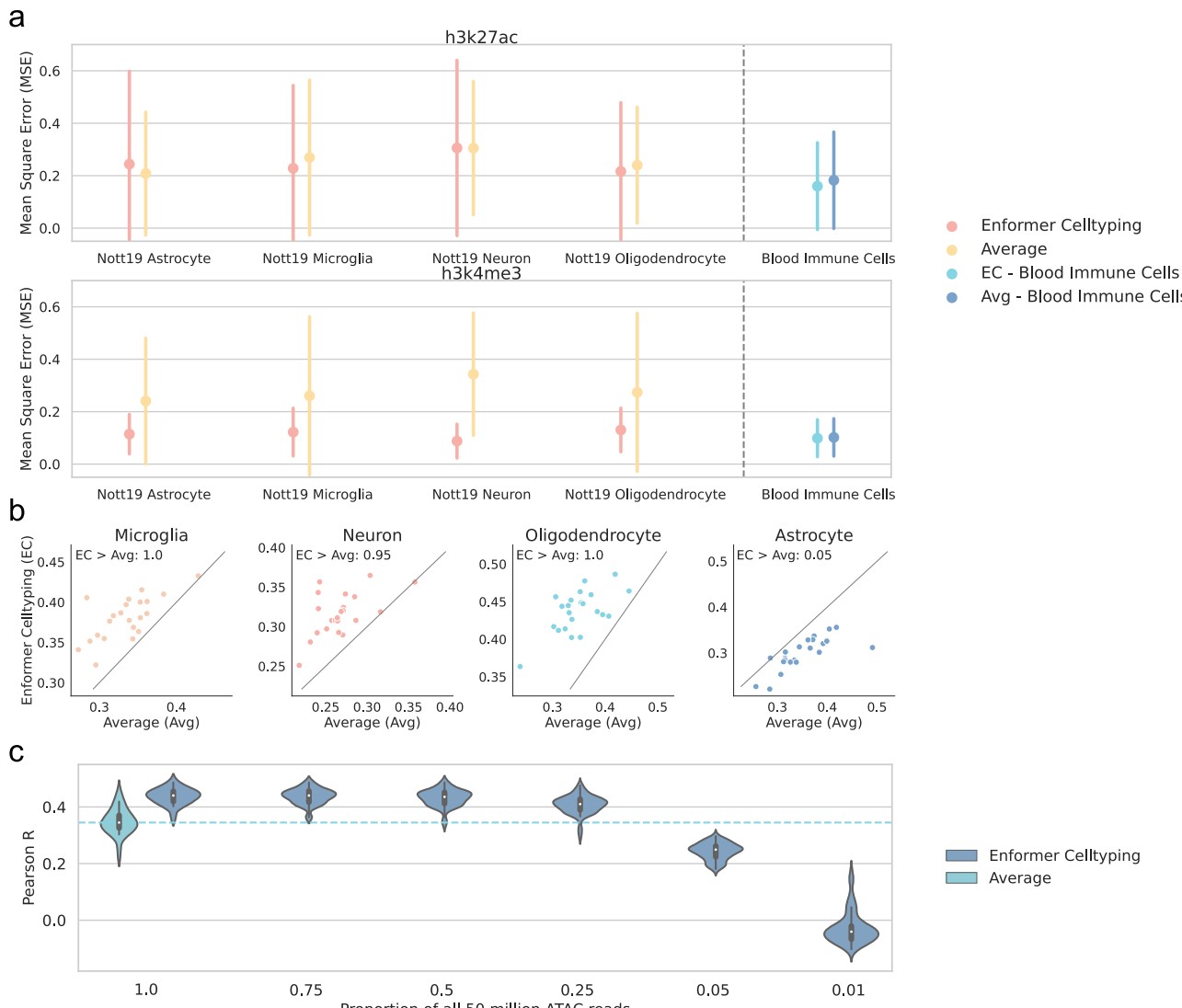

**Fig. 5 | Enformer Celltyping can predict in cell types generated outside of ENCODE. a** Enformer Celltyping's mean squared error (MSE) across four isolated cortical brain cell types from Nott et al.[27]. The average performance for Enformer Celltyping and the average signal across the three ENCODE immune cell types (Blood Immune Cells) is also shown for comparison. The results for the "Blood immune cells" is the union of prediction results for T-cells, Neutrophils and Monocytes. "Average" here refers to the MSE calculated using the average values for the histone mark, across all cell types used in the training of Enformer Cell-typing. Performance is measured for H3K27ac and H3K4me3 which were both assayed in the study. Performance is based on genome-wide predictions (*n* = 8086 genomic loci for each histone mark, cell type and model combination) with error bars for the standard deviation. **b** Enformer Celltyping's Pearson R correlation with the true signal (y-axis) compared against the Average's (avg) Pearson R (y-axis) in distal regulatory regions sourced from Search Candidate cis-Regulatory Elements by ENCODE (SCREEN)[26] and derived from a diverse range of ENCODE cell types. Performance was measured for H3K27ac which has a known functional role in this region. Pearson R values were calculated per chromosome for each cell type (representing a dot in the figure). **c** Pearson R correlation (y-axis) on distal regulatory regions from SCREEN for Oligodendrocytes at differing proportions of the 50 million reads (x-axis). Subsampling was repeated 5 times with different seeds. Median of violin plots were −0.04, 0.249, 0.410, 0.436, 0.440 and 0.440 for proportions 0.01, 0.05, 0.25, 0.5, 0.75, 1 for Enformer Celltyping and 0.345 for Average at proportion of 1. Source data are provided as a Source Data file.

measured histone marks should give a fair indication for any future applications of the model (Fig. 5a, b). However, we did process the data and derive the -log10 *p*-value probability of binding using the same ENCODE pipeline to make it comparable. To enable a fair comparison between ENCODE and non-ENCODE cell types, we show performance against the three ENCODE immune cell types which were held out of the Enformer Celltyping training dataset: grouped and they are referred to as "Blood immune cells". Overall, Enformer Celltyping's performance did not worsen drastically (relative to the ENCODE blood immune cells) despite the cells' dissimilarity to ENCODE samples. This becomes more impressive when we consider the performance of the average signal which showed a sharp increase in the MSE for H3K4me3 in the non-ENCODE cell types, relative to the ENCODE blood immune

cells. Enformer Celltyping's ability to predict well in the presence of such a domain shift compared to the average signal of the ENCODE training cell types, highlights its ability to use the cell type-specific chromatin accessibility effectively when making predictions. Moreover, Enformer Celltyping outperforms the average signal at distal enhancer regulatory regions in three of the four cell types in at least 95% of the cell type-chromosome comparisons (Fig. 5b), highlighting its superior performance and utility predicting in functionally relevant, cell type-specific regions. The model did underperform for astrocytes which we believe was, at least partly, contributed to by the lower ATAC-Seq sequencing depth, at approximately 36 million across the two replicates compared to the 50 million read depth Enformer Celltyping was trained on. To investigate how chromatin accessibility (ATAC-Seq)

data sequencing depth, an input for Enformer Celltyping, impacts model performance more generally, we down-sampled the ATAC signal at varying levels 5 times with differing seeds on oligodendrocytes (Fig. 5c). Even at a sequencing depth of 50% of what the model was trained to predict with, Enformer Celltyping outperforms the average signal in the cell type-specific, distal enhancer regions, highlighting the model's applicability under data quality variations.

To further test Enformer Celltyping's robustness and broader applicability, we considered cancer research, validating its genome-wide predictive performance on ENCODE cancer cell lines which were not part of the training data (Supplementary Fig. 7a, b). The model showed no deterioration in its predictive quality compared to the immune cells lines (Supplementary Fig. 7a). Moreover, Enformer Celltyping outperformed the average signal at all regulatory regions, especially in enhancer regions, known to be cell type-specific; with better performance in 91% of proximal and 100% of distal enhancer cell type-chromosome comparisons (Supplementary Fig. 7b). These results highlight Enformer Celltyping's efficacy at predicting histone mark signals across varied cellular environments. Moreover, given its strong predictive performance at enhancer regions (Fig. 4a, Supplementary Fig. 7b) and to investigate the usability of Enformer Celltyping's embeddings, these features were used with a random forest classifier to distinguish super-enhancers from other enhancer regions in cancer cell lines. Super-enhancers are enhancer regions with a multitude of TF binding sites, often occupied by master transcription factors/coactivators like MED1 and BRD4 which are fundamental to cell type-identity[28]. We found these super-enhancers to be highly cell type-specific (Supplementary Fig. 7c), so a method to predict cell type-specific super-enhancers would be useful, highlighting the benefit of Enformer Celltyping's ability to predict in previously unseen cell types. Enformer Celltyping's embeddings showed strong performance on these held-out cell lines (mean area under ROC of 0.85) (Supplementary Fig. 7d). Finally, Enformer Celltyping's performance across all 9 cell types tested is highlighted in Supplementary Fig. 9, demonstrating the model's consistent performance across these diverse cell types.

## Predicting genetic variant effects on the epigenome

Having validated Enformer Celltyping's ability to predict the cell type-specific, genome-wide histone mark signals conditioned with chromatin accessibility information, we proceeded to test its primary application of predicting the cell type-specific effect of non-coding, disease-relevant genetic variants on the epigenome. The requirements of this task are why Enformer Celltyping was designed in the manner that it was: able to predict in novel cell types (such that it can predict SNP effects in disease relevant cells); as well as incorporating the effect of distal genetic regulators (which we showed to be necessary to capture the effect of genetic variants in Supplementary Fig. 2).

We developed an framework, building on past approaches utilising QTL datasets[16,29,30], systematically measuring the genome-wide correlation between our model's predictions and the measured effects from the QTL studies using SLDP[30] (see Methods section Histone Mark QTL SLDP and Data collection and processing). We used QTL datasets as they directly measure the effect of extant human genetic variation. Furthermore, we have not concentrated on predicting the effect of genetic variants in isolation due to the confounding effect of linkage disequilibrium (LD), obscuring the coinherited and casual SNPs[30]. Approaches like fine-mapping could help identify the casual SNPs, however recent works have highlighted that despite considerable advancements there is still opportunity for improvement[31–33]. Note that past work concentrated on using eQTL studies and predicting genetic variant effects in the same cell types that the models were trained on. On the other hand, we utilised hQTL datasets and our model's ability to predict in previously unseen cell types using their chromatin accessibility data, matching the cell types studied in the hQTL to gain insight into the model's predictive ability of SNP effects on the epigenome. Moreover, we implemented strict pre-processing criteria, filtering out any interactions from the hQTL set where the distance was greater than the model's receptive field and measured the effect across the full receptive field of the model.

We used hQTL data from the Blueprint project phase 2[10], for two histone marks (H3K4me1 and H3K27ac), each of which was obtained for three cell types (T-cells, Monocytes and Neutrophils). These were the same cell types which were held out of the Enformer Celltyping training data, as discussed above. These hQTL datasets were generated using whole genome sequencing from neutrophils ($n = 196$), monocytes ($n = 194$) and T-cells ($n = 169$) individuals. We note that while these are the largest published hQTL datasets, they remain small by the standards of modern eQTL datasets, and so could be considered underpowered to detect QTL effects. We tested 867,568 genetic effect predictions for all of Enformer Celltyping's predicted histone marks for six cell types against the six hQTL datasets[10]. Similarly to model training, we tested the effect on the forward, reverse strand and with small random shifts in genomic starting position to return the average effect. We expected to see a significant result for the prediction in the same cell type and histone mark for which the hQTL study was completed (Fig. 6a).

The results of the SLDP genetic effect predictions were worse than what was expected with only one of six of the matching histone mark and cell types being nominally significant and none after false discovery rate (FDR) correction of 0.05 (Fig. 6a). We noted that one possible causes of differences, relates not to the quality of SNP effect predictions, but to differences in predictions of peaks for the reference sequence: sometimes Enformer Celltyping doesn't predict peaks which are seen in the experimental data. If a peak is missed in the predictions, then the effect of a SNP which decreases the size of that (missing) peak, cannot be captured. To assess whether this affects the results, we tried filtering out SNP-to-histone QTLs associated with peaks that were missed by Enformer Celltyping, and re-running the SLDP analysis. However, this did not improve the significant matched cell type and mark cases (Supplementary Fig. 9).

We next tested whether the inability to replicate the hQTL SNP effect signals are specific to Enformer Celltyping or is more generally a trait of genomic deep learning models trained using variation across the genome (which has been noted by other recent work[34]). We reran our framework on the same hQTL sets using the matched cell type and histone mark output tracks from Enformer (Fig. 6b). Interestingly, Enformer similarly fails to capture the hQTL SNP effect signals with only one expected significant result for H3K27ac in T-Cells. This finding highlights how the current training paradigm of genomic deep learning models is insufficient to capture the effect of genetic variants sufficiently with or without extrapolating predictions to previously unseen cell types.

In making these predictions, we altered our approach to in silico mutagenesis, relative to the standard in the field (Supplementary Fig. 10). One issue with the standard approach, which centres the SNP in the input window, is that it does not utilise the full receptive field of these models (Supplementary Fig. 10a, b). Our approach captures the full receptive field by making multiple predictions for each SNP, sliding the input window (Supplementary Fig. 10c). Given our approach results in multiple predictions of the same genomic locations, we sought to confirm that predictions were consistent for the same genomic region, regardless of where it appears in the input window (Fig. 6c) (also see Methods section Off-centre correlation analysis). This resulted in extremely high correlations for all tested cell types, confirming that centring the model's predictions on the SNP and shifting the input window in the hQTL dataset would not affect the histone mark binding predictions.

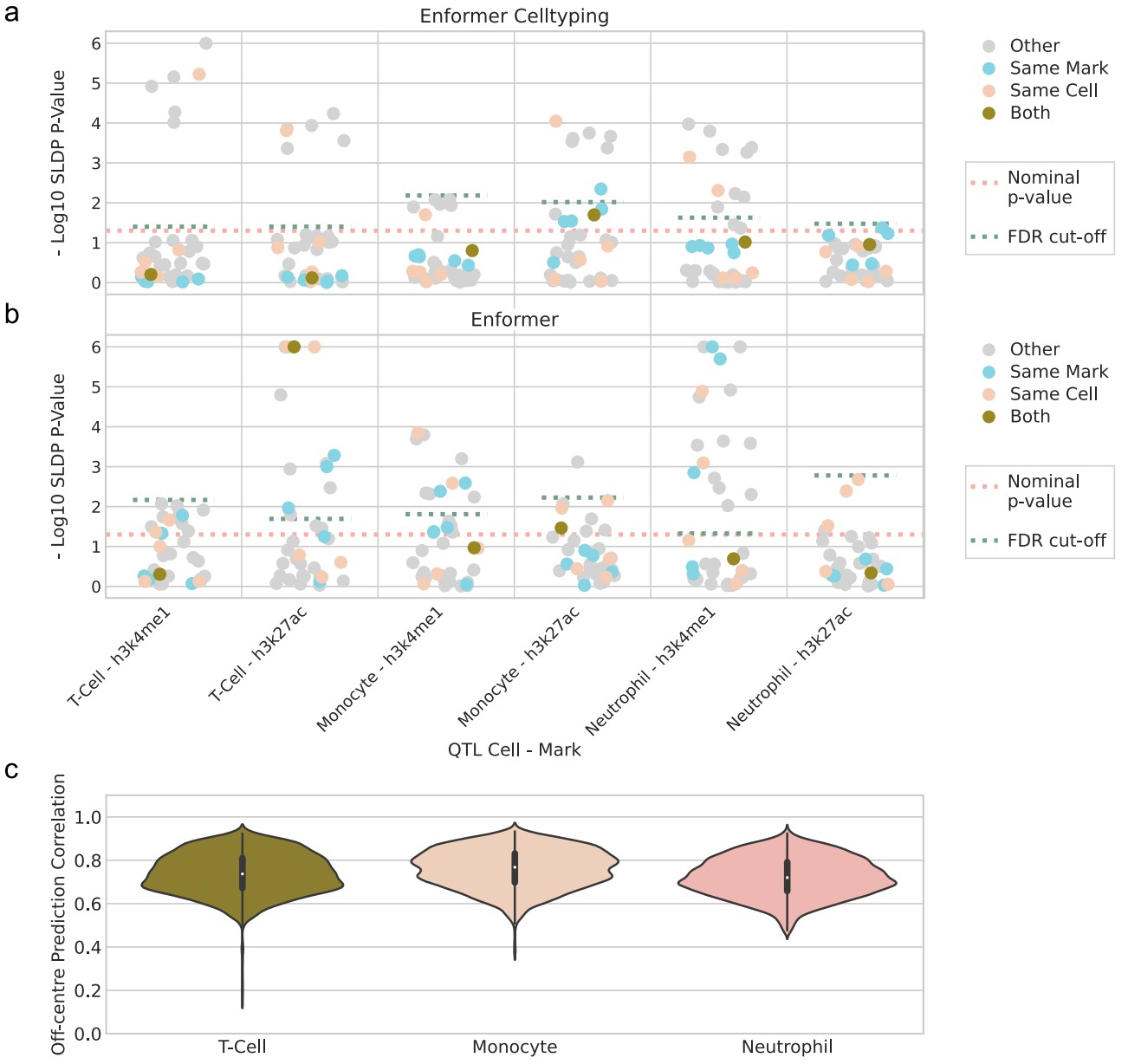

**Fig. 6 | Genomic deep learning models' genetic effect predictions predominately do not replicate QTL results.** Statistical significance (y-axis) of SLDP[30] genome-wide concordance between Enformer Celltyping's (**a**) or Enformer's (**b**) genetic variant predictions and measured hQTL effect sizes. The x-axis depicts the six Blueprint phase 2 hQTL studies[10] and the cell-histone mark predictions are coloured by their relationship to the hQTL study: 'Both' – both the histone mark and the cell predicted in were the same as the hQTL study, 'Same Mark' – the mark matched, 'Same Cell' – the cell matched and 'Other' – neither the cell nor mark matched. If there was perfect agreement between our model and the hQTL studies, all 'Both' entries would be significant. A nominal *p*-value from SLDP (pink dashed line) and the cut-off after Benjamini and Hochberg correction based on the minimum, non-significant *p*-value, corrected at the level of each hQTL (Turquoise dashed line) are both added. **c** Enformer Celltyping makes consistent predictions for a genomic region, regardless of where it lies within the model's receptive field. The correlation (y-axis) in Enformer Celltyping's histone mark predictions at matching genomic regions when the model's input window is moved. This was tested for the three immune cell types (x-axis) for 1000 genomic positions each with and without genetic variants added. Source data are provided as a Source Data file.

## Investigating cell type-specificity of Enformer Celltyping predictions

We have demonstrated that Enformer Celltyping accounts for changes in local chromatin accessibility for its predictions (Fig. 1c) but also that it considers global, cell type-specific signals (Figs. 2 and 5). To investigate what was driving the model's cell type-specificity, we identified the genomic regions with distal acting histone mark peaks that had the highest contribution from the global chromatin accessibility signal for a selection of cell types (Fig. 7a and Methods section Global chromatin accessibility signal – cell type-specific motif enrichment). The corresponding DNA sequences for these peaks were assessed for

enrichment of known motifs using Homer[35] (Fig. 7b, c). Finally for each cell type, based on the set of significant known motifs, we looked for cell type specificity of the related transcription factors using EWCE[36] (Fig. 7d).

Overall, Fig. 7 highlights how Enformer Celltyping identifies DNA motifs relating to distinct transcription factors for the different cell types tested and some of these transcription factors match the same cell types in their transcriptional specificity. For example, monocytes had significant enrichment in myeloid cells. Furthermore, the strongest enrichment for the heart, which contains epithelial cells in the epicardium, had the largest enrichment in corneal epithelial cells.

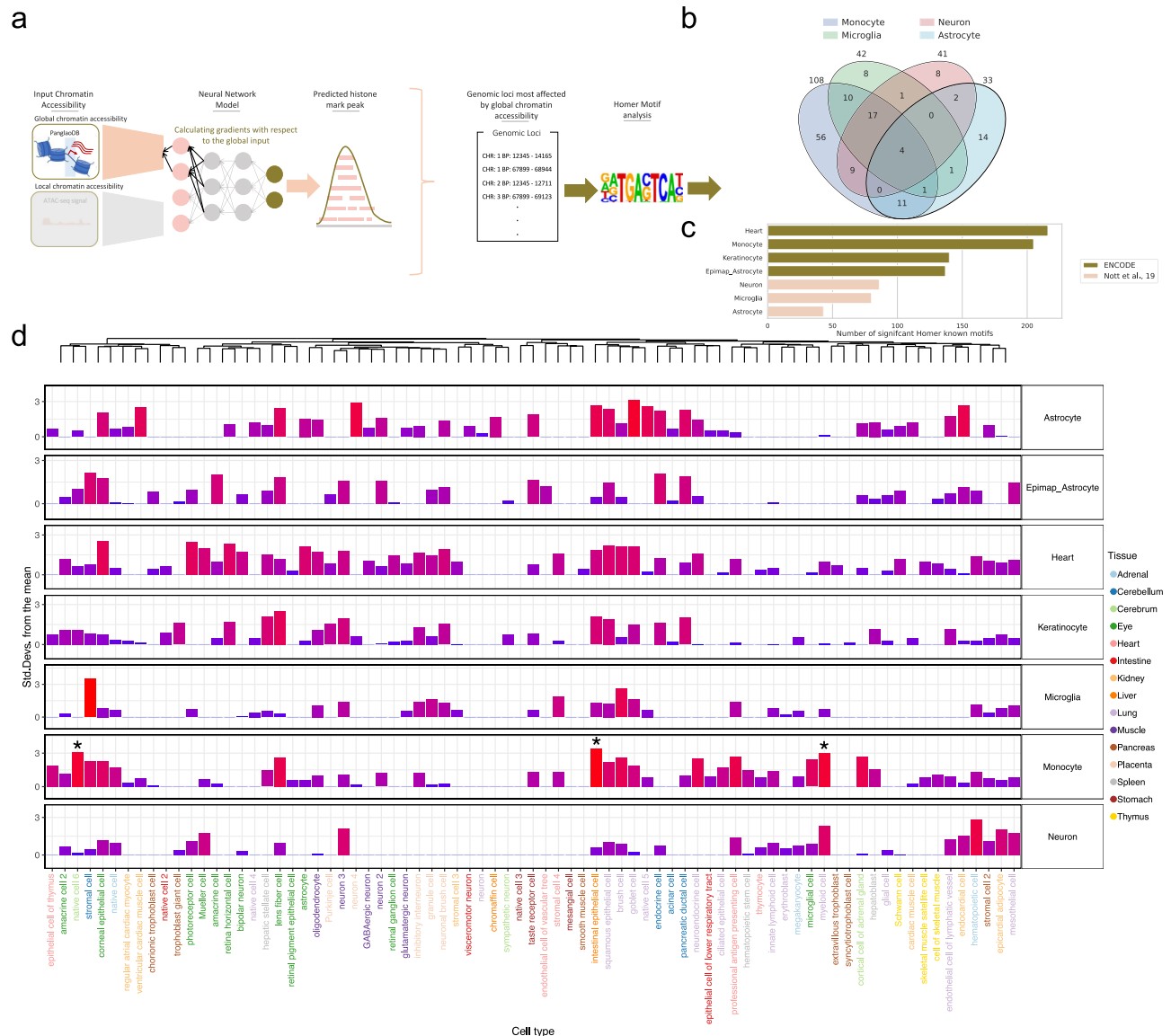

**Fig. 7 | Enformer Celltyping identifies motifs for cell type-specific transcription factors which are enriched based on the global chromatin accessibility signal.** **a** For each peak, the influence of the global signal was approximated by calculating the partial derivatives of the model with respect to the input, i.e. the gradient on the input. From this, peaks and their genomic loci that rely the most on the global input signal were identified. These top loci were analysed with Homer[35] for each cell type. **b** The overlap between significant transcription factors across the immune and brain cell types tested for all histone marks. **c** The number of significant transcription factors per cell type. **d** Cell type-specificity of the transcription factors from Homer for each cell type, calculated using EWCE[36]. The rows are the differ cell types for which predictions were made – note the brain cell types are taken from Nott et al.[27] with the exception of Epimap_Astrocyte which was from ENCODE[17] sample available through EpiMap[19]. The top row is a dendrogram displaying the clustering of cell types from the Descartes human whole body[62] single-cell RNA–seq reference dataset used. The columns, ordered by the dendrogram, are the cell types tested from the reference dataset with the colour representing the tissue of origin. Significant associations found by EWCE (false discovery rate adjusted *p*-value < 0.05), indicating enrichment in a cell type are marked with an asterisk. The background geneset were the approximately 400 transcription factor genes from Homer. Source data are provided as a Source Data file.

Neurons showed strong enrichment in neuronal cell types (neuron 3). Keratinocytes had the largest enrichment in lens fibre cells which have been noted to resemble keratin-rich keratinocytes in the stratum corneum of the epidermis and share biological processes including organelle degradation during their terminal differentiation[37]. The differentiated astrocyte cell from ENCODE[17] (Epimap_Astrocyte) and the isolated astrocyte from Nott et al.[27], showed large cell type enrichments in brain cell types, in neurons 3 and 4 respectively but not in the reference astrocyte cell type. Moreover, we see overlap in the transcription factors identified with Homer in similar cell types such as across the immune cells; microglia and monocytes (Fig. 7b). Interestingly, the cell types from Nott et al.[27] (Astrocyte, Microglia and Neurons) showed found fewer transcription factors from the global

chromatin dependent peak motifs (Fig. 7c). This could indicate a bias in Enformer Celltyping whereby, although the domain shift of predicting in isolated, single cell types outside of ENCODE does not seem to affect the genome-wide predictive performance, it does affect the global cell type-identity the model learns. We can also see this effect when we consider the projection of the model's embedded global chromatin accessibility signal for the Nott et al. brain cell types versus the ENCODE brain cell types which seem to separate in a lower dimensional space (Supplementary Fig. 11).

## Leveraging Enformer Celltyping for complex trait associations
Finally, Enformer Celltyping showed strong performance in distal regulatory regions (Figs. 4c, 5b, Supplementary Fig. 7b) so, despite

having limited success at accurately capturing the effect of genetic variants (Fig. 6), we queried whether the predicted enhancer regions were enriched for genetic variants associated with complex traits and diseases in cell type-specific regulatory regions (Fig. 8).

We first aimed to confirm that we could recapitulate the genetic enrichment results of Nott et al.[27]; despite using the same dataset, to ensure comparability with our predictions, we had to omit samples and alter their analysis. Specifically, our predictions are continuous,

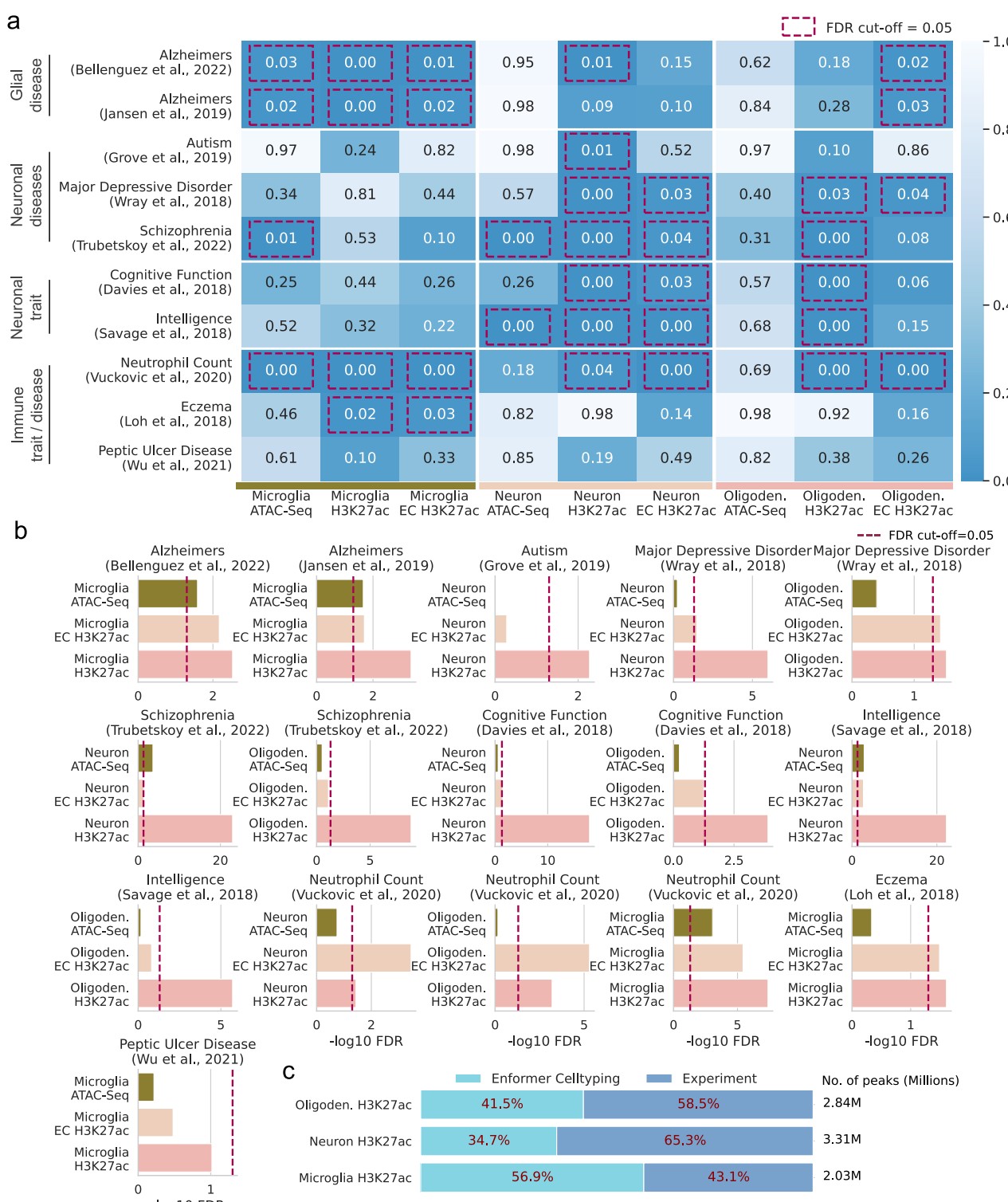

**Fig. 8 | Enformer Celltyping's predictions capture cell type-specific genetic enrichment for complex traits. a** Heatmap of stratified LD score regression (s-LDSC)[73] analysis for genetic variants associated with brain and immune diseases/traits and behavioural traits (sourced from associated GWAS) displayed as false discovery rate (FDR) value for significance of enrichment for ATAC-Seq chromatin accessibility signal, H3K27ac signal and Enformer Celltyping's (EC) predictions of H3K27ac for microglia, neurons and oligodendrocytes (oligoden.). **b** -log₁₀(FDR) genetic enrichment for the complex traits from s-LDSC. **c** Proportion of peaks in derived peak files used for s-LDSC analysis. The median, minima and maxima foe the violin plots were Monocyte; 0.768, 0.380, 0.934, Neutrophil; 0.720, 0.475, 0.925 and T-Cell; 0.737, 0.159, 0.924. Source data are provided as a Source Data file.

genome-wide signals (i.e. in bigwig format) whereas we ran s-LDSC with binary annotations (i.e. peaks in bed format): To ensure the real and predicted signals were comparable, they were uniformly processed. Using the experimental H3K27ac signal, we recapitulated Nott et al.'s results, finding known associations in glial and neuronal diseases and traits like microglia with Alzheimer's disease, neurons with schizophrenia and neurons with intelligence (Fig. 8a). Moreover, we included immune GWAS traits and diseases (eczema and neutrophil count) and found enrichments for microglia, the resident macrophage of the brain. Given this, we assumed the significant associations for the experimental H3K27ac signals were true. Thus, we could next test whether EC's predictions, which use ATAC-Seq chromatin accessibility data, improve the capture of genetic variants for these known trait-to-celltype relationships over the enrichment seen in ATAC-Seq.

Enformer Celltyping's predictions captured the vast majority of known enrichments with significant associations (Fig. 8a). Furthermore, the predictions improved on the ATAC-Seq data in 14 of the 16 expected enrichments (Fig. 8b). It is worth noting that for some cases the level of enrichment of the predicted H3K27ac signal did not match the experimental H3K27ac (Fig. 8b). This was most notable for neurons in Schizophrenia and Major depressive disorder. We believe this was, at least in part, due to the relatively lower number of peaks in the derived bed files for the predictions compared to the experimental data (Fig. 8c). Overall, these results show how Enformer Celltyping can be used along with the breadth of publicly available ATAC-Seq data to derive histone mark signals which harbour greater enrichment for genetic variants associated with complex traits and diseases than chromatin accessibility alone, highlighting it's utility for the study of complex traits and disease.

## Discussion

We report Enformer Celltyping, a deep learning model to predict epigenetic signals in previously unseen cell types, using DNA sequence and chromatin accessibility data from the cell type of interest to make its predictions, with the largest receptive field to date of 100,000 base-pairs. Enformer Celltyping achieves this receptive field through a transfer learning approach based on Enformer, a model which predicts gene expression and epigenetic signals limited to the same cell types it was trained on[16]. Here, we recreated Enformer; removing the output and convolutional layers after the transformer block and freezing the weights in pre-trained layers. This approach differs from past applications which mostly fit a linear model on top of Enformer's output[34,38], offering more flexibility and aligning with current approaches in the machine learning domain such as NLP[39]. Our approach shows the power of transfer learning using large, pre-trained models in computational biology. Here, we trained Enformer Celltyping on a data set several times larger than Enformer's, using a fraction of computational resources (132 versus 5376 GPU hours). We believe this approach, which we have made openly available, shows promise for what researchers can achieve fine-tuning large, pre-trained models in the face of limited resources.

Enformer Celltyping achieves best-in-class performance for histone mark predictions (Fig. 3a), strong predictive performance in functional genomic regulatory regions (Fig. 4) and maintains accuracy even with a domain shift to isolated cell types assayed outside of ENCODE (Fig. 5). Developed as a quantitative model, Enformer Celltyping predicts a continuous -$\log_{10}$ $p$-value signal, which has been shown to yield better generalisation and interpretability than binary, classification models[40]. Moreover, the -$\log_{10}$ $p$-value derived after MACS 2.0 peak calling was chosen over the fold change measure as it typically has a higher signal-to-noise ratio[20,41]. However, there are some limitations to both our approach to model training and Enformer Celltyping's performance. Firstly, due to a lack of availability, we used imputed chromatin accessibility ATAC-Seq data for the majority of cell types from EpiMap[19]. A possible improvement could be to use assayed

chromatin accessibility data or use a better performing imputation technique[20,42]. Our reliance here on imputed data highlights the importance of generating new biological datasets to supplement those already available which have been paramount for the training genomic deep learning models like ENCODE. Secondly, Enformer Celltyping's predictions are made in 128 base-pair bins. This is lower resolution than the standard bin size after peak calling from ENCODE (25 base-pairs)[17]. A major goal of developing Enformer Celltyping was to use it to predict the effect of genetic variants. Doing so at a lower resolution will dilute the effective change on the epigenetic signal which may have contributed to the suboptimal performance of Enformer Celltyping on the genetic variant analysis. Predicting in 25 base-pair bins was not possible as a result of the transfer learning approach using the frozen layers from Enformer. Possible solutions to enable this would be to either allow the weights to update in Enformer so it can adjust to the desired resolution which would incur a much higher a computational cost, equivalent to that which the original authors faced or to use a U-Net architecture on top of Enformer to increase the resolution, as proposed recently[43]. Furthermore, upon investigating the causes of the model's predictions based on global cell type information, we found that cell-type specific histone mark predictions are influenced by the presence of cell type-specific transcription factor motifs. However, there were only a limited number of expected significant enrichments, which could be due to confounders such as the limited number of transcription factors with known motifs. Moreover, mapping these back to human based on orthologs and the fact that some motifs will span whole transcription factor families may affect the quality of the results.

A primary goal of machine learning models that predict gene expression or epigenetic signals from DNA, including Enformer Celltyping, is to predict the effect of genetic variants[16,29,30]. Usually, these models can only make predictions in the same cell types they were trained in. Here, we have expanded this utility to predict in any cell type based on its chromatin accessibility data, enabling genetic variant predictions in any cell type of interest which is more aligned with researcher's needs for investigation into disease-relevant cell types. For use with our model and for the wider community, we created the first framework for genetic variant effect evaluation using hQTL data in combination with SLDP[30]. Previous studies have evaluated performance using eQTL data, which is several layers further removed from the presumed direct biological effect of SNPs (eQTLs aggregate over multiple layers of gene regulation, while hQTLs will be more directly affected). Moreover, other studies have used fine-mapping to evaluate model performance, but such approaches are compromised by their reliance on fine-mapping approaches, which differ depending on the methodology used[31–33]. Our approach utilises a model's full receptive field (Supplementary Fig. 10), along with interpretable $p$-value cut-offs to measure associations, as opposed to only considering z-scores as previously considered[16,30]. This framework can be used to evaluate any future genomic deep learning models as the field focuses on improving their performance in genetic variant effect predictions.

Overall, Enformer Celltyping's performance in predicting the effect of genetic variants predominately did not replicate the QTL results using the SLDP measure of concordance[30] (Fig. 6a). We hypothesised that two issues contributed to this: Firstly, that the hQTL set contained interactions which caused a histone mark binding position to be removed by a genetic variant but these histone mark binding positions were not captured in Enformer Celltyping's prediction on the major allele. Secondly, that genomic deep learning models, including Enformer Celltyping, inherently struggle to accurately predict the effect of genetic variants based on their current training paradigm. We tested the effect of the first issue by removing such interactions before running SLDP but this had little effect on increasing the number of expected significant associations (Supplementary Fig. 9). The second issue has already received growing attention in the literature,

specifically in relation to Enformer. Firstly, Karollus et al.[38] showed how the predicted effect of regulatory elements decreased with distance and did not match the measured causal effects of these elements, such as enhancers. This means that on average, Enformer struggles to identify distal regulatory genomic regions, weighting their importance less than local regulatory information. This may give insight into Enformer Celltyping's relatively poor performance in predicting the effect of genetic variants where the majority of genetic variants tested were distal (Supplementary Fig. 2). Secondly, Sasse et al.[34] comprehensively showed how Enformer underperforms at predicting the effect of genetic variants on transcription, even predicting the incorrect direction of effect in up to 40% of tested cases. The authors hypothesised that this was a result of Enformer being trained on points across the reference genome and then needing to make out-of-sample predictions for individuals' personal genetic variants at any locus. In our work, these issues could be compounded by the fact that models like Enformer Celltyping, which predict the effect of genetic variants in previously unseen cell types, need alternative functional genomic information to infer the cell type it is predicting in. For Enformer Celltyping, this is chromatin accessibility information. However, this chromatin accessibility data used for the genetic variant effect predictions did not account for the genetic variant thus further disadvantaging the model's ability to predict the genetic effect accurately. In our work, we also tested the extent to which this affected our results by running the same genetic variant effect prediction test on Enformer (Fig. 6b). However, Enformer performed similarly highlighting that it did not play a major role in the results. We believe the real limitation is the lack of genetic variation seen in the current approach to genomic deep learning model's training which we have noted for epigenetics and which was highlighted for transcription by Sasse et al. Therefore, we agree with Sasse et al.[34] that future work hoping to use a machine learning model to predict the effect of genetic variants should concentrate on training models on data which accounts for genetic variants and secondly, that research should focus on predicting in the same cell types upon which the model was trained rather than extrapolating to previously unseen cell types using functional genomic data from the cell type of interest like chromatin accessibility information. This is, of course, limited by the current lack of suitable public data, specifically for epigenetics. Despite genomic deep learning model's inability to predict the effect of genetic variants, our results highlight Enformer Celltyping's utility for complex trait research; the predicted histone mark signals harbour greater enrichment for genetic variants associated with complex traits and diseases than chromatin accessibility signals (Fig. 8).

In summary, current genomic deep learning models like Enformer are not suitable to study the epigenome and the effect of genetic variants on disease-relevant, cell or sub-cell types not captured in public repositories. Here, we introduce Enformer Celltyping, a model which can predict epigenetic signals in previously unseen cell types, using a large receptive field to capture genomic regulatory information. Enformer Celltyping requires just chromatin accessibility data from the cell type of interest to enable predictions of regulatory effects on six histone mark signals. Moreover, we introduce a framework to benchmark methods for genetic variant effect predictions and show how the training paradigm for current genomic deep learning models is insufficient to accurately capture these genetic variant effects. Despite this, we show Enformer Celltyping's ability to capture genetic enrichment of complex traits.

## Methods

### Data collection and processing

The training data was derived from GRCh37 DNA sequence and p-value continuous tracks after peak calling in EpiMap[19], encompassing data from the ENCODE project[17]. The p-value was used over fold change down to its greater signal-to-noise ratio (see Supplementary Note for more information). The DNA sequence was one-hot encoded following the same approach as previous models[12,13,15,16]. The aim for the histone mark tracks data collection from EpiMap was to maximise both the number of histone marks the model could predict and also the number of cell types which due to the sparse nature of ENCODE data, results in a trade-off between the two (Supplementary Fig. 12). Any cell types with assays marked as low quality by EpiMap were removed from the training set. The final training set included ChIP-seq histone mark data for H3K27ac, H3K4me1, H3K4me3, H3K9me3, H3K27me3 and H3K36me3 from 104 cell types (Supplementary Table 1). Given the current preference of ATAC-Seq over DNase-Seq to measure chromatin accessibility (ATAC-Seq has similar sensitivity and specificity to DNase-Seq with three to five times fewer cells[44]), we opted to train our model using ATAC-Seq data so it would be more applicable to future users. To avoid losing cell types, imputed chromatin accessibility ATAC-Seq data sourced from EpiMap[19] was used for any where there was not an observed track which was the case for the majority (98 out of 104) of cell types. All tracks were converted from 25 base-pair average signals to 128 base-pair averaging to match the resolution of the Enformer[16]. All training data tracks measure the -$\log_{10}$ adjusted p-value from MACS2 peak calling[45], indicating the statistical significance of a genomic position i.e. is the site likely to be a real binding site when compared against a background. The -log10 p-value derived after MACS 2.0 peak calling was chosen over the fold change measure as it typically has a higher signal-to-noise ratio[41]. This is the same reason that ENCODE chose -log10 p-value as the primary signal tracks for analyses[46]. Moreover, -log10 p-values are more interpretable which is useful for downstream analysis of predictions, for example thresholding for peaks for functional regions done with ChromHMM[47]. Such thresholding of output can not be easily done with fold change values.

Despite EpiMap's uniform reprocessing, to avoid possible issues like sequencing depth biases, we followed previous approaches of training on the arcsinh-transformed signal[20,48,49]:

$$\sinh^{-1}x = \ln(x + \sqrt{1+x^2})$$

We achieved clear separation across epigenetic marks on our training cell types after this transformation which was not present otherwise (Supplementary Fig. 13). The average chromatin accessibility and histone mark signals, used for model training, were derived by averaging these 104 cell types' tracks (see Methods section Enformer Celltyping architecture).

Two separate sets of cells were used as test cases for the model; 3 immune cell types sourced from EpiMap[19] (CD14+ monocytes, CD16+ neutrophils and naive CD4+ T cells) and Nott et al.'s[27] isolated cell types from resected cortical brain tissue (PU.1+ microglia, NeuN+ neuronal, OLIG2+ oligodendrocyte and NeuN- LHX2+ astrocytes). The Nott et al. data was processed to produce consensus peak p-value tracks across samples (different individuals were taken as separate replicates) using the ENCODE ChIP-Seq and ATAC-Seq pipelines (https://github.com/ENCODE-DCC/chip-seq-pipeline2 and https://github.com/ENCODE-DCC/atac-seq-pipeline) with the supplied conda environment and default parameters. Note, this was the same processing that was conducted on the training data. One exception to the default parameters was subsampling the signals to 30 million reads for the ChIP-Seq and 50 million for the ATAC-Seq to match the processing of the EpiMap data and avoiding performance loss due to sequencing depth issues[50]. Identically to the training cell types, the test sets were averaged to 128 base-pair resolution and arcsinh-transformed. However, none of these test cell types or any similar cell types, such as other immune cells, from EpiMap were used in the training set. Moreover, these were not used to derive the average histone mark or chromatin accessibility tracks to avoid data leakage between training and test sets[50].

The histone QTL (hQTL) Blueprint phase 2 data[10] was downloaded from the European Genome-Phenome Archive (ID: EGAD00001005199).

The six hQTL sets (CD14+ monocytes, CD16+ neutrophils and naive CD4+ T cells for histone marks H3K4me1 and H3K27ac) were uniformly processed using MungeSumstats (v1.6.0)[51], using the format_sumstats function and default parameters with the exception of the check_dups parameter which was set to false to enable quality control and formatting of QTL datasets. To transform the hQTL datasets to be compared with Enformer Celltyping's SNP predictions, all SNP to histone mark peak interactions further than Enformer Celltyping's receptive field, or any trans interactions, were removed along with all SNPs not included in HapMap3[30]. SNPs effects were then aggregated to a single value per SNP using the following transformation, as done in previous work[16,29,30]:

$$\widehat{\alpha_m} = \frac{1}{\sqrt{|G_m|}} \sum_{k \in G_m} \hat{\alpha}_m^{(k)}$$

Where $G_m$ is the set of all histone mark peaks for which a cis-hQTL test was performed for variant $m$ and $\hat{\alpha}_m^{(k)}$ is the marginal correlation of SNP $m$ and histone mark peak $k$'s signal[30] (i.e. the Beta effect value from the study).

### Calculating genetic variant distance from epigenetic effect peak in QTL associations

The distance from the SNP to the associated histone mark peak for the six histone mark hQTL BLUEPRINT Phase 2 study[10] were aggregated to one genetic variant per LD block to avoid double counting. Aggregation was done to keep SNPs with the lowest $p$-value for each LD block and if multiple with the same, the shortest distance to the effect region (to avoid biases towards distal interactions). The Pickrell LD blocks[52] (available from https://bitbucket.org/nygcresearch/ldetect-data) were used for this analysis (Supplementary Fig. 2). This highlights the need for models which can account for these distal interactions when predicting epigenetic profiles, not only transcriptional profiles.

### Training data

Enformer Celltyping was trained on 67,007 GRCh37 genomic regions across chromosomes 1–22 and across the 104 training cell types. The regions were picked by first binning the genome based on the size of the model's predictive window (114,688 base-pairs). These bins were then filtered at a DNA level and also a cell type level. The DNA filters were to exclude bins in ENCODE blacklist regions[17] and ensuring subsequent genomic positions up and downstream of the region so the model can utilise the full input window size (196,608 base-pairs). The cell type level filters were to ensure a coverage of histone mark signals > 12.5% in the region (based on a -log$_{10}$ $p$-value cut-off of 2 to mark a peak[19] for a mark in a 128 base-pair bin). This was done for each histone mark independently and then all histone mark regions were downsampled to match that of the histone mark with the lowest number of peaks, resulting in 11,903 positions for each histone mark. This approach ensured equal representation and avoided the model prioritising training on a subset of over-represented marks[50]. The filtering resulted in 67,007 genomic regions across all training cell types and corresponds to 14,188 unique DNA positions which were split into a training and validation set by chromosome, a similar number of unique genomic regions to past approaches[13,16]. The forward and reverse complement sequence and small, random sequence shifts up and downstream for each were also generated for each genomic region to be included in the training set. The validation set was created by splitting 20% of the training cell types and the start position for these was randomly permuted around the chosen regions to avoid the model overfitting to the training bins. A list of the training positions and cell types is available for download through figshare (https://figshare.com/projects/Enformer_Celltyping/159143).

For each position in each histone mark, the average and distribution of arcsinh-transformed, -log$_{10}$ $p$-values across the training cell types were calculated for the pre-training step (see Methods

section Enformer Celltyping architecture). This distribution was created by binning signals for each histone mark and genomic region, based on their -log$_{10}$ $p$-value signal into 10 groups at 0.5 intervals from 0 to 4.5. The distributions for each histone mark were calculated before model training, the code for which (and all other data processing and model training/testing) is available in our Github repository (https://github.com/neurogenomics/EnformerCelltyping)[23].

### Pre-trained Enformer model

Enformer Celltyping uses a pre-trained version of Enformer[16] which has had its pre-trained layers removed after the multi-headed attention layers and its weights frozen (Supplementary Fig. 1). A custom approach was developed to be able to apply transfer learning to Enformer. The Enformer model available from Tensorflow Hub has not been formatted to allow modifying of its layers or unfreezing weights (it doesn't expose a _call_ method, or any of its internal variables, and lacks a model signature). To circumvent this issue, we first extracted the weight matrices from the available, trained version of Enformer and rebuilt an untrained version of Enformer following the architecture provided in the original study[16]. We next manually mapped layers across the two models and once completed, updated the weights in each layer of the untrained Enformer model to the values of the trained version of Enformer. Once complete, we tested the predictions against the trained version of Enformer to ensure the weights were mapped correctly. From here, we removed the final layers after the multi-headed attention layers and froze the weights of the other layers and incorporated this pre-trained, chopped version of Enformer into our Enformer Celltyping model. This recreated version of Enformer is now fully customisable allowing flexibility in the transfer learning approach such as fine-tuning through Tensorflow API. We have made an automated script to create this version of Enformer available in our Github repository (https://github.com/neurogenomics/EnformerCelltyping)[23] so others can similarly use Enformer in their own problem domains.

### Enformer celltyping architecture

Enformer Celltyping was implemented in Tensorflow v2.4 with Sonnet v2 functionality in the pre-trained Enformer model. Enformer Celltyping was trained with a two-step approach – a pre-training step with a separate training of DNA sequence and chromatin accessibility submodules, and a subsequent cell type-specific training step, combining the output of the two model branches with subsequent layers. The pre-training step improved model performance overall (Supplementary Fig. 4).

For the pre-training, Enformer Celltyping's architecture is split into two separate submodules (Supplementary Fig. 1a), both with their own optimiser and loss function. The first "DNA" submodule uses the DNA sequence to predict the average and distribution of each histone mark. The second "celltyping" submodule predicts the difference between the average histone mark signal and the cell type-specific signal. These are explained in more detail below; how the data is constructed is discussed in Methods section Training data. The output blocks of the first DNA submodule is shown in Supplementary Fig. 1a as "Output avg" and "Output distribution" while the second celltyping submodule's pretraining output is shown as "Output delta". It is the first "DNA" submodule which uses the pre-trained, chopped, frozen version of Enformer[16] through a transfer learning approach (discussed above in "Pre-trained Enformer model").

The first is a "DNA" submodule takes one-hot encoded DNA sequence of length 196,608 base-pairs (A = [1,0,0,0], C = [0,1,0,0], G = [0,0,1,0], T = [0,0,0,1]) as its input, and processes it through four main parts: (1) seven convolutional blocks with pooling, (2) eleven transformer blocks, and (3) a cropping layer and (4) a further dense layer and convolutional block and output layer. During both training steps, the layers corresponding to the first three parts of the architecture (the pre-trained Enformer layers) were frozen and only the final part's

weights were updated through backpropagation. The DNA submodule outputs an average cell type prediction and distribution across the training cell types for each histone mark at the genomic region. The average cell type -log10 $p$-value score for each histone mark, used to train the model, was derived from the 104 EpiMap, training cell types (see Methods section Data collection and processing). The distribution data corresponds to the proportion of training cells whose histone mark, arcsinh-transformed -$\log_{10}$ $p$-value fall into each of 10 bins between 0 and >=4.5 at 0.5 intervals for a given genomic location. For example, if for H3K27ac signal at a genomic location, all cell types have no signal i.e. -$\log_{10}$ $p$-value = 0, then the distribution to predict would be [**1**,0,0,0,0,0,0,0,0,0]. Whereas, if two cell types of the 104 had -$\log_{10}$ $p$-value = 5.5, then the distribution would be [**.98**,0,0,0,0,0,0,0,0,**0.019**]. Including the distribution of histone mark signals as well as the average in the DNA module prediction for the pre-training step helps capture variability of regions across cell types and improved the overall performance of Enformer Celltyping (Supplementary Fig. 5).

The second "celltyping" submodule takes two inputs; a local and global representation of chromatin accessibility (ATAC-Seq) for the cell type. The local chromatin accessibility is the ATAC-Seq from the 199,936 base-pairs averaged at 128 base-pair resolution of the same genomic location as the DNA (1562 positions). This local chromatin accessibility signal is first pre-processed by calculating the difference between the cell type-specific signal and the average chromatin accessibility signal of the training before passing to the model. Whereas, the global chromatin accessibility corresponds to the chromatin accessibility for 3000 base-pairs around the transcriptional start site of 1216 marker genes, averaged at 250 base-pair resolution (3.648 million base-pairs, input size of 14,592 positions). These marker genes were taken from a database of all known human marker genes derived from over 1000 single-cell RNA-Seq experiments for any cell type, collated by PangloaDB[53]. Human marker genes were excluded if they fell outside of chromosomes 1–22 or in ENCODE blacklist regions[17], leaving 1216 genes. The list of PangloaDB genes and code to derive the global chromatin accessibility is available at: https://github.com/neurogenomics/EnformerCelltyping[23]. We tested the performance of using each of the global and local chromatin accessibility signals separately and combined for a subset of 300 training steps, highlighting the benefit of incorporating both (Supplementary Fig. 3). Note that the global chromatin accessibility signal will be the same for a cell type regardless of the genomic region it is predicting in. Similar to Avocado's approach to capture differing information about the epigenetic landscape[20] and more broadly, other approaches in the field of NLP[54], the global and local chromatin accessibility information were embedded at different resolutions. These embeddings are then flattened, concatenated and passed through two dense layers before being passed to a separate output head per histone mark. Note that this flattening step after the embedding layers does not affect the positional information of the chromatin accessibility information. The celltyping submodule outputs the difference between the average histone mark prediction and the cell type of interest for the genomic region.

For the full training step, Enformer Celltyping combines the three outputs; "Output avg", "Output distribution" and "Output delta", of the "DNA" and "celltyping" submodules with subsequent convolutional and dense layers (Supplementary Fig. 1b) to predict the cell type-specific histone mark signal for six histone mark (H3K27ac, H3K4me1, H3K4me3, H3K9me3, H3K27me3 and H3K36me3), in a given genomic region. Enformer Celltyping's output has six channels, one for each histone mark and 896 positions, corresponding to the centre 114,688 base-pairs of the input window, aggregated into 128 base-pair resolution bins. Only the centre 114,688 base-pairs are predicted on to avoid predicting on the edge positions which do not have as many neighbouring positions upon which to make a prediction[16]. Enformer

Celltyping is trained to predict arcsinh-transformed -$\log_{10}$ $p$-value, however, this can be converted back to a raw -$\log_{10}$ $p$-value which Enformer Celltyping outputs by default.

## Enformer celltyping training

Enformer Celltyping was trained using Tensorflow API for 1000 steps of a batch size of 128 cell type-specific genomic positions (128,000 positions) as a pre-training stage taking approximately 4 days (see Methods section Enformer Celltyping architecture). The combined architecture, full training stage was run for 4 full epochs (6940 steps with a batch size of 128 cell type-specific genomic positions equating to 888,320 positions) stopping when the model started to overfit on the validation dataset, taking approximately 1.5 days on a Nvidia A100, 80GB RAM GPU. A learning rate of 0.0002 was used for pre-training and 0.005 for the full training stage and the ADAM optimiser[55] was used for both. The low initial learning rate was chosen to allow for learning rate warmup and avoid early overfitting[56] after which we matched the learning rate to similar models[14,16]. In the pre-training stage, the loss function differed across the submodules; a Poisson negative log-likelihood loss function was used for the average signal prediction (the same as past approaches[13,16]) and cross-entropy loss was used for the distribution in the DNA submodule whereas a mean squared error (MSE) loss function was used for the celltyping submodule following other epigenetic embedding approaches[5,20], given the possibility of negative values. The Poisson negative log-likelihood loss function was used for the full training stage. Since Enformer Celltyping freezes the weights relating to the chopped, pre-trained Enformer model (see Methods section Pre-trained Enformer model), the output from these layers for each genomic position was computed just once for each genomic location and then load as needed for each epoch. As a result of this, training was substantially quicker than for Enformer in the original study[16] despite our model seeing more positions for multiple epochs compared to Enformer's single epoch approach, totalling 132 GPU hours for pre and full training stages versus 5376 GPU hours.

## Testing receptive field

To test the receptive field of both Enformer and Enformer Celltyping, we evaluated the effect of random genetic variants on histone mark predictions across increasing distances (from 0 to 98,304 base-pairs up and downstream). Specifically, we measured the average change in prediction across reference and alternative sequence on the centre four positions (512 base-pairs) of the output signal. DNA sequences were simulated by randomly sampling base-pairs. Next, a single random genetic variant per simulated DNA sequence was inserted and the effect on prediction measured. The location of the genetic variants were spread across 1000 evenly spaced locations across the 196,608 base-pair input. This random sequence and random variant simulation was repeated 100 times for each genetic variant location (i.e. 100,000 simulations overall) to capture the average effect of any variant. All six output channels of Enformer Celltyping were averaged whereas only the output channels from Enformer corresponding to histone mark signals were considered. The same random iteration seed was used for both models so that the same DNA sequences and genetic variants would be simulated. The average change across reference and alternative was shown at the position of the genetic variant (Fig. 1b).

Similarly, we tested the effect of changes in the local chromatin accessibility on Enformer Celltyping by measuring the average change in prediction across reference and alternative chromatin accessibility signal on the centre four positions (512 base-pairs) of the output signal. We replaced the cell type-specific local chromatin accessibility signal of 640 base-pairs with the average signal at increasing distances up to the full receptive field of the model (Fig. 1c). The code for testing the DNA and chromatin accessibility receptive fields are available at: https://github.com/neurogenomics/EnformerCelltyping[23].

## Visualising cell type embedding

A side-effect of embedding the global chromatin accessibility information in Enformer Celltyping is that these latent representations of cell types is continuously improved by the model during back-propagation. We can then visualise these latent representations with a 2D projection. or all 104 training cell types in our training set we plotted the global chromatin accessibility embeddings using UMAP[57] with default parameters (Fig. 2a).

## Benchmarking

Enformer Celltyping was benchmarked against Epitome[18] across the six histone marks for three immune cell types from EpiMap[19] (CD14+ monocytes, CD16+ neutrophils and naive CD4+ T cells) (Fig. 3a). Epitome predicts transcription factor binding and histone marks for a cell type using chromatin accessibility (DNase or ATAC-Seq) data and was shown to have best-in-class predictive performance at cell type-specific histone mark predictions[18].

The weights for a pre-trained version of Epitome, trained on Panc1, PC-9, OCI-LY7, MCF-7, Karpas-422, IMR-90, HepG2, HeLa-S3, HCT116, H9, H1, GM23338, GM23248, GM12878, A549 cells from ENCODE[17], were supplied by the authors and which we have made available through figshare (https://figshare.com/projects/Enformer_Celltyping/159143). Both Epitome and Enformer Celltyping were used to make genome-wide predictions of the same six histone marks for the three immune cell types. Epitome predicts in 200 base-pair bins in a classification setting – whether there is a peak in a 200 base-pair interval or not whereas Enformer Celltyping has been developed to predict signal values directly (-$\log_{10}$ $p$-values) at 128 base-pair resolution. To compare the methods, Enformer Celltyping's predictions were converted into classification scores using a -$\log_{10}$ $p$-value cut-off of 2, similar to previous work[19]. The predictions for the two models were averaged to the lowest common multiple between the two (3200 base-pairs). Both of these differences led to a blurring of signal and sub-optimal predictions for both approaches but were necessary for comparison. As used in previous work[20], the performance was evaluated using a balanced measure of performance, accounting for the disparity in the number of regions with and without peaks, the average-precision (AP) for each cell type – histone mark combination. AP is a single score that represents the precision-recall curve based on the rank of predictions using the mean of precisions for each threshold, with the increase in recall from the previous threshold considered as the weight.

We did not benchmark Enformer Celltyping against Enformer as they perform different, non-comparable tasks - Enformer predicts in held-out chromosomes and so does not have to extrapolate predictions to previously unseen cell types whereas Enformer Celltyping does on whole, held-out cell types (see Supplementary Note for more information).

## Super-enhancer prediction in cancer cell lines

To determine how informative Enformer Celltyping's embeddings are given the strong performance at enhancer regions (Fig. 4c), we trained a random forest classifier to distinguish super-enhancers from normal enhancers in cancer cell lines, using four-fold cross-validation, holding out all data from one cell line in each fold. We used the scikit-learn 1.0.2 implementation[58] with default parameters, expect setting the maximum features considered per decision tree split to $\log_2$ of the total number of features due to the large input size. Super-enhancers are enhancer regions with a multitude of transcription factor binding sites, often identified by occupation of high levels of master transcription factors and coactivators[59,60]. Super-enhancer regions were obtained from the dbSuper[28] database for the cell lines MCF-7, K562, HEPG2 and GM12878. While the identification of all enhancer regions was from their ENCODE H3K27ac bed files MCF-7 (https://www.encodeproject.org/files/ENCFF024TEH/@@download/ENCFF024TEH.bed.gz), K562

(https://www.encodeproject.org/files/ENCFF044JNJ/@@download/ENCFF044JNJ.bed.gz), HEPG2 (https://www.encodeproject.org/files/ENCFF392KDI/@@download/ENCFF392KDI.bed.gz) and GM12878 (https://www.encodeproject.org/files/ENCFF367KIF/@@download/ENCFF367KIF.bed.gz). We down-sampled the number of enhancer regions (negative samples) to match the number of super-enhancers (positive samples) in the cell line and visualised performance on the held-out test cell line with receiver operating characteristics (ROC) curves.

## Off-centre correlation analysis

Enformer Celltyping makes histone mark predictions for 128 base-pair bins in a 114,688 base-pairs wide predictive window. We tested whether sliding the genomic location, i.e. the DNA and local chromatin accessibility information, would result in a change in the histone mark prediction at a given genomic location. We tested this by randomly sampling 1000 genomic positions from our three immune cell types (CD14+ monocytes, CD16+ neutrophils and naive CD4+ T cells) which have SNP information in the Blueprint phase 2 hQTL studies data[10]. For each position, we measured the correlation in matching genomic regions between the centred prediction and two off-centre predictions, one up and one down-stream. The positions up and down-stream where chosen to capture the full input window as output (Supplementary Fig. 10), to capture as much overlap as possible. Finally we reported the range of these correlations for both genomic data with and without a SNP to ensure genetic variants did not cause a bias in the results (Fig. 6c). The code for testing the off-centre correlation analysis is available at: https://github.com/neurogenomics/EnformerCelltyping[23].

## Histone mark QTL SLDP

Although multiple approaches have now been developed for in silico mutagenesis, there is little consensus on how best to benchmark performance. Experimental datasets including CRISPR, MPRAs and xQTL mapping studies have all been proposed[16,34,38]. However, currently only xQTL mapping studies can be carried out in vivo on large sample sizes. Thousands of genetic variants are tested in one study based on the natural variation in the population. However, associations detected in xQTL mapping studies are confounded by linkage disequilibrium (LD), where alleles are co-inherited based on their physical proximity[22], making it difficult to identify the causal SNPs[30]. Therefore, LD needs to be taken into account when comparing these datasets against model predictions.

For our benchmarking framework, we predicted the effect of each genetic variant from the six hQTL datasets[10] with Enformer Celltyping by predicting the histone mark signal for both the reference and the alternative sequence, calculating their difference and summing this effect across the full genomic window to get a single, signed score for each SNP. Despite our model predicting the measured change in histone mark binding across the full receptive field, aggregating the effect to a single value is necessary to match the format of the hQTL data and for SLDP. We predicted in six cell types, three matching immune cells and three unrelated cell type from EpiMap: Pancreas, Leg Muscle and Stomach tissue, to act as negative controls. To obtain a prediction across the full input window size, and thus utilising the full receptive field of Enformer Celltyping, we predicted with the SNP first centred in the input window and then performed two further predictions with the SNP slid off-centre (Supplementary Fig. 10) and appended these three output windows before calculating the score. Finally, we averaged scores computed using the forward and reverse complement sequence and with small, random sequence shifts up and downstream. This meant for each SNP, we predicted (2x reference and alt, 3x full receptive field, 4x reverse complement and random shift) 24 predictions. We predicted for 867,568 SNPs across the six hQTL datasets for the six cell types, resulting in approximately 125 million predictions.

We utilised signed linkage disequilibrium profile regression (SLDP)[30] to measure the statistical concordance between the signed variant effects (our model's predictions) and the genome-wide association study's marginal correlations (the aggregated hQTL SNP values, see Methods section Data collection and processing). SLDP uses generalized least-squares regression to measure the agreement between these, iteratively inverting the direction of the signed variant effect measures along with their neighbouring entries in blocks to derive a null distribution. The method accounts for population linkage disequilibrium (LD) so does not require any fine-mapping strategies. The measured agreement defines how important the variants are to the phenotype's heritability[30]. We ran SLDP for all combinations of our cell type predictions and histone mark with each hQTL set (6x cell types, 6x histone mark predictions and 6x hQTL sets, resulting in 216 tests) and report the results in Fig. 6a. All code to run the genetic effect predictions and SLDP analysis (with a conda environment) is available at: https://github.com/neurogenomics/EnformerCelltyping[23].

### Global chromatin accessibility signal – cell type-specific motif enrichment

We inspected the cell type-specific motifs based on the global chromatin accessibility signal to infer what was driving the cell type-specific signal the model captures. We first predicted genome-wide tracks for histone marks acting outside of the TSS or gene body (H3K27ac, H3K4me1 and H3K9me3), as these would be more likely to harbour cell type-specific transcription factors, using Enformer Celltyping and identified peaks in the data with -log10 *p*-value score >2, at 1024 base pair resolution. For each peak, the influence of the global signal was approximated by calculating the partial derivatives of the model with respect to the input, i.e. the gradient on the input[61]. The results for each peak was ordered by absolute value and the top 10% of peaks reliant on the global signal were identified. The DNA at these positions were run through Homer's[35] known motif analysis to search for motif enrichment using default settings. The Homer motif analysis is available at: https://github.com/neurogenomics/EnformerCelltyping[23].

The transcription factors' genes relating to the resulting motifs, using an FDR cut-off of 0.05 for the Homer known motif results, were tested for cell type specificity using EWCE[36] bootstrapping (repeated 10,000 times with a false discovery rate (FDR) adjusted *p* value threshold of 0.05). All known motifs from Homer were used as the background set (approximately 400 transcription factor genes) after mapping non-human genes to human based on one-to-one orthologs and any motifs overlapping all 7 cell types were removed to get the cell type-specific motifs (4 in total). The single-cell RNA-Seq reference dataset used to determine cell type enrichment was the Descartes human whole body, fetal dataset containing approximately 377,000 cells and 77 distinct cell types[62]. This approach was repeated for all cell types of interest and FDR multiple test correction was implemented to account for the repeated tests (Fig. 7).

### Cell type-specific genetic enrichment for complex traits

To test for genetic enrichment of complex traits, Enformer Celltyping's H3k27ac predictions for Nott et al.'s[27] microglia, neurons and oligodendrocytes (continuous signals) were converted to bedGraph and finally to bed files, using a peak cut-off of arcsinh(1) for inclusion in the bed file; this led to better enrichment than more stringent peak cut-offs. To enable a fair comparison, the ATAC-Seq and experimental H3K27ac signals for the same cell types were converted using the same approach.

Summary statistics for genome-wide association studies (GWAS) for glial diseases - Alzheimer's[63,64], neuronal diseases - autism spectrum disorder[65], major depressive disorder[66] and schizophrenia[67], neuronal traits – cognitive function[68] and intelligence[69] and immune diseases/traits – neutrophil count[70], eczema[71] and peptic ulcer disease[72] were downloaded and uniformly processed with MungeSumstats v1.11.3[51]

(default settings, converting build to hg19 where necessary and saving in 'LDSC' format).

We applied stratified LD score regression (s-LDSC)[73] v1.0.1 (https://github.com/bulik/ldsc) to test for disease enrichment. Specifically, annotation files for each of the signals for each cell type of genomic loci were first created with Phase 3 of the 1000 genomes reference. Followed by the generation of LD scores with a window size of 1 centiMorgan (cM) i.e. approximately 1 million base pairs, filtering to HapMap3 SNPs to match the baseline model. Finally, the enrichment analysis was run for the GWAS summary statistics across the different cell types and signals as well as those in the baseline model whilst excluding the major histocompatibility complex (MHC) (due to the known difficulties predicting LD in this region)[73]. All code to process the data and run the s-LDSC analysis (with conda environments) is available at: https://github.com/neurogenomics/EnformerCelltyping[23].

### Reporting summary
Further information on research design is available in the Nature Portfolio Reporting Summary linked to this article.

## Data availability
All training and test cell types from EpiMap, along with scripts to download and complete all pre-processing steps are available at https://github.com/neurogenomics/EnformerCelltyping[23]. Blueprint phase 2, histone mark QTL data was downloaded from the European Genome-Phenome Archive (ID: EGAD00001005199). The full training and all validation scripts are also available at https://github.com/neurogenomics/EnformerCelltyping[23]. A list of the training regions and cell types, the one-hot encoded DNA sequence, the trained Enformer Celltyping model's weights, the average chromatin accessibility signal of the training cell types and the SNP effect predictions on the hQTL sets for both Enformer and Enformer Celltyping are all available through figshare (https://figshare.com/projects/Enformer_Celltyping/159143). For the interest of the reviewer and readers, we have also made available full genome browser track visualisations for T-Cells H3K27ac, including the experimental track, Enformer Celltyping and average predictions: https://genome-euro.ucsc.edu/s/almurphy/EC_hg19. Source data are provided with this paper.

## Code availability
The Enformer Celltyping model architecture, training scripts and all analysis are available from https://github.com/neurogenomics/EnformerCelltyping[23].

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

## Acknowledgements

I would like to thank Leo Linden, Simon Mathis, David Kelley and Jacob Schreiber for their invaluable conversations and feedback that supported this work. I would also like to thank Alyssa Morrow for her help supplying a version of Epitome for benchmarking. This study makes use of data generated by the Blueprint Consortium. A full list of the investigators who contributed to the generation of the data is available from www.blueprint-epigenome.eu. Funding for the project was provided by the European Union's Seventh Framework Programme (FP7/2007-2013) under grant agreement no 282510 – BLUEPRINT. This work is supported by the UK Dementia Research Institute award number UK DRI-5008 through UK DRI Ltd, principally funded by the UK Medical Research Council. N.S. also received funding from a UKRI Future Leaders Fellowship [grant number MR/T04327X/1].

## Author contributions

A.E.M. and N.G.S. jointly conceived of and executed the study. W.B. wrote initial scripts to extract weights from the Enformer tensorflow model. M.R. and M.P. provided guidance on metalogical approach. A.E.M wrote the manuscript which was reviewed by N.G.S., M.P. and W.B.

## Competing interests

The authors declare no competing interests.
