## [Peer Review file · Nature Communications]

Predicting cell type-specific epigenomic profiles accounting for distal genetic effects

Corresponding Author: Mr Alan Murphy

Version 0:

Reviewer comments:

Reviewer #1

(Remarks to the Author)

This manuscript presents a computational work called Enformer Celltyping to predict histone mark activity for previously unseen cell types. Overall, I think this work is interesting and useful, however, both its advantages and disadvantages of this work is obvious. My main concerns are listed:

1. The manuscript is very comprehensive, with a comprehensive data curation and the benchmark. However, some description is tedious. For example, the introduction part is too lengthiness and I can not quickly get the point of the work. I strongly suggest the author to highlight its core point, i.e., the prediction of epigenetic marks for "unseen" cell types. Also the discussion part is messy, which will cover up the merit of the work.
2. The main disadvantages of the work for me is the methodology part, which is somewhat not innovative and not clear. It seems that the authors applied a foundation model (pre-trained model) Enformer (it seems to be developed by other group) and fine-tuned it to the novel cell types for epigenetics mark prediction. Also I can not quite get the point that for a totally novel cell type, if there are not any training data available for this cell type, How the model can be fine-tuned? It seems to be a zero-shot learning scenario and a route fine-tuning strategy is not applicable here. I strongly suggest the authors to reformulate the description of the methodology part to clarify this point.
3. The limitation of the work, as the authors discussed, is clear that it can not accurately predict genetic variant effect.

Reviewer #2

(Remarks to the Author)

In this paper, the authors present "Enformer Celltyping," a deep learning model designed to assess the impact of genetic variants on the epigenome. This model stands out for its ability to predict epigenetic signals irrespective of cell type and considers distal DNA interactions up to 100,000 base pairs away, an improvement over traditional method. Enformer Celltyping demonstrates more performance compared to existing approaches, showing some robust results across various cell types and biological regions, even those not included in the initial training. The model also includes an application in regulatory quantitative trait loci mapping, highlighting its broad utility in genomic studies. Using a custom transfer learning approach derived from a pre-trained Enformer model, the system efficiently addresses the distal effects of genetic codes on regulatory signals, saving significant memory and training time. Additionally, Enformer Celltyping incorporates both genome-wide and local representations of cell types based on chromatin accessibility, adopting techniques from NLP to those used in the Avocado. This integration makes the model applicable to other cell types with available accessibility data. Validation of Enformer Celltyping has been carried out against leading methods in various biological settings. Overall, this paper represents a contribution to the field of genomics, addressing the challenge of profiling epigenetic effects across diverse cell types. The model's ability to predict epigenetic signals independently of cell type and to consider distal DNA interactions marks an advancement.

However, this manuscript could be strengthened by addressing some concerns. Firstly, to further substantiate the model's robustness and broad applicability, it would be beneficial to observe its performance on additional, well-characterized cell

lines within the ENCODE project, such as the MCF-7 breast cancer line. Testing the model on MCF-7 would not only confirm its efficacy across varied cellular environments but also underscore its utility in cancer research—a field profoundly influenced by epigenetic modifications. Moreover, given the model's demonstrated success in predicting enhancer activities, extending its application to super-enhancers in the same MCF-7 cell line could be significant.

Reviewer #3

(Remarks to the Author)

In this manuscript, Murphy et al presented Enformer Celltyping, a model that predicts histone marks by incorporating a recent sequence-based model, Enformer, and cell type-specific ATAC-seq features. Additionally, the authors used *in silico* mutagenesis to identify histone QTLs. The model architecture design presents a new way to combine pre-trained sequence-based model with cell-type-specific finetuning steps. Although the model improved the cell type feature beyond the Enformer model, and outperforms a baseline Epitome, the model still seems to lack sufficient accuracy for predicting histone modifications, and lacks performance to recapitulate hQTLs from experiments.

Main comments:

1. Enformer has histone ChIP-seq as training data. Examining the representative tracks in Figure 1, it seems Enformer Celltyping prediction have both false positives and false negatives. Does the finetuned model improve over the original Enformer? If so, how much does the performance improvements of Enformer Celltyping over Epitome (and original Enformer) come from the original Enformer component?
2. The main evaluation metric is MSE. However, MSE may not be a good metric because it is affected by data scale. Correlation metric like Pearson/Spearman correlation at bin level will be more informative. Moreover, general performance evaluation across a few independent samples, especially focusing on the cell-type-specific genomic regions, will better present the model performance. One way to evaluate is to merge all peak loci from both experimental data and predicted data across multiple cell types, and evaluate the performance in these differential group.
3. In Figure 2, was the microglia peak validated in experiments? Plotting experimental tracks across the samples and compare them with predictions will help better evaluate the model performance.
5. In Figure 4 the No Peak region, it seems neutrophil and T-cell-specific predictions have higher MSE than average. Why is that? Does it indicate bad prediction quality in the background, meaning more false positives? The authors can show a confusion matrix heatmap for every cell type so that people can better understand the performance difference between cell types.
6. When performing mutagenesis, did the authors modify only the DNA sequence without changing ATAC-seq signal? That would be a problem because the ATAC signal was likely to change as a result of the sequence change. In that case, the model could output invalid predictions.
7. When masking global accessibility signal to evaluate model's reliance on the global accessibility, did the author train a separate model (without global accessibility) or use the original model but masking it during prediction? If using the original model, it can produce invalid predictions as well since the masking process was unseen during training.
8. In Figure 7, there are many missing hits, and the predicted results does not seem to be interpretable. For example, in monocyte group, the highest hit is unexpectedly categorized as astrocyte. The other screening results also seemed very mixed. How to understand these results and do they make sense?

Technical comments:

1. Does predicting p-value make sense? The distribution of p value is weird and can interfere with training process. One alternative is \log_2fc .
2. How does ATAC-seq data quality affects prediction? How does subsampling the bam files for varied depth affects prediction?
3. In Sup Fig 1a when flatten the cell type embeddings, how was it performed? Was the position information affected in this process?
4. Is the p-val calculation in Nott et al. the same as Encode pipeline? Otherwise it would be difficult to justify model performance across databases.

Minor issue: 884: "ffect" Typo.

Reviewer #4

(Remarks to the Author)

Version 1:

Reviewer comments:

Reviewer #1

(Remarks to the Author)

I have no more question technically.

Reviewer #2

(Remarks to the Author)

The authors address my concerns accordingly.

Reviewer #3

(Remarks to the Author)

The reviewer appreciate the authors' efforts in responding the comments and concerns in depth, and is fully aware of the importance of cell-type-specific predictions. However, the reviewer is worried that core issues/concerns, particularly regarding model accuracy, was not clearly addressed, which would not make the model truly achieve the expected function for predicting cell type-specific epigenetic signals.

1. Concerns on the model's performance/accuracy. It does not seem the authors addressed the concerns regarding the inaccuracy of the model prediction. The authors agreed that the Enformer Celltyping's performance is similar to the average signal of the training dataset, noticing the very minimal difference between "Enformer Celltyping" and "Avg" groups in Figure 3b. Then the authors claimed that "Enformer Celltyping's cell type-specific predictions (Enformer Celltyping) outperform that of the average predictions from pre-training (EC Avg) in the vast majority of cell types and histone marks", so that to support that "Enformer Celltyping picks up on cell type-specific signals". It is misleading to claim in this way, as the data only has a minimal difference, similar to the evidence between "Enformer Celltyping" and "Avg" groups mentioned above. It is important to report the confusing metrics and simultaneously to examine the actual performance when compared to the experimental data. The reviewer appreciate the authors adding experimental data tracks during the revision in Figure 2b. Examining the data at the AIF1 locus (in Figure 1d and 2b together), it is clear that the model predicts the best in constitutive 'active' regions – regions where there are minimal cell-type-specific signals. However, the Enformer Celltyping model did not correctly capture the expected signals at AIF1 promoter or upstream enhancer regions – false positives and false negatives from the current model prediction are very evident in these regions, and to some extent seems randomly assigned.

2. how much does the current model improve over the basal model, the original Enformer? Since the Enformer parallelly predicts tracks over many cell types included in the training. The reviewer's original suggestion of comparison to Enformer can easily be performed by comparing the model performance in the Enformer training cell types (which were also used as training cell types in the current Enformer Celltyping model). In cell types that are both included as training data, it is expected that the current model will perform much better – if the current model truly works – than the original Enformer, especially in the regions where Enformer does not perform well.

Other comments:

The reviewer does not agree with the claim that ENCODE data was not able to train good machine learning models. Many good ML models have been trained with ENCODE data and achieved great performance. It may require more new data for the many more ML training, but the reviewer believes that the current ENCODE data are overall of good and standardized quality for ML applications.

There are still many typos in the manuscript. Some of them from the discussion section: Lines 939, 962, 965,

Reviewer #4

(Remarks to the Author)

Version 2:

Reviewer comments:

Reviewer #3

(Remarks to the Author)

I thank the authors for the detailed explanation, but with all respect to the authors and their continuous efforts, I still hold my

concerns about the model's performance/accuracy from the last round of review. I believe that a new ML model (or any other new technology) should provide significant advancement to the field. However, I do not see this model achieving that level of advancement to predict cell-type-SPECIFIC epigenome features beyond the original Enformer model. To elaborate on it, I attached a file, showing the screenshot of Figure 2b in the manuscript, and in comparison, to only show the cell-type-specific regions together with the provided experimental data (by masking cell-type-shared regions and cell types that do not have accompanied experimental data for validation). The remaining results clearly show that the current model predicts 'cell-type-specific' epigenetic signals in an almost random way. Thus I do not believe the current model would make meaningful contribution to real-world applications beyond the original Enformer model.

Reviewers' comments:

Reviewer #1 (Remarks to the Author):

This manuscript presents a computational work called Enformer Celltyping to predict histone mark activity for previously unseen cell types. Overall, I think this work is interesting and useful, however, both its advantages and disadvantages of this work is obvious. My main concerns are listed:

1.1 The manuscript is very comprehensive, with a comprehensive data curation and the benchmark. However, some description is tedious. For example, the introduction part is too lengthiness and I can not quickly get the point of the work. I strongly suggest the author to highlight its core point, i.e., the prediction of epigenetic marks for "unseen" cell types.

1.1 We thank the reviewer for this and their other notes on the manuscript's text. On reflection, we believe we tried to cover too many aspects of such models (and field more generally) and have now majorly restructured our work to concentrate on its main points. To this end, we have shortened the introduction section by a third and made the main point of the work more clear: Enformer Celltyping enables epigenetic predictions in previously unseen cell types by using DNA and chromatin accessibility information and secondly, the development of a new framework of measuring the effect of genetic variants and genetic enrichment for complex traits for such models. We would kindly ask the reviewer to read the updated version to ensure it is comprehensible and succinct.

1.2 Also the discussion part is messy, which will cover up the merit of the work.

1.2 Similar to the previous point, we agree that the discussion section was not clear. We have now shortened it by approximately a third and made the main point of the work more clear. For example, see the final summarising paragraph of the discussion (line 1044). To note, we have also updated the abstract, results and conclusion to similarly reflect the main points of the paper which will hopefully further aid in comprehension. We invite the reviewer to similarly reread these sections to ensure they are clear.

2.1 The main disadvantages of the work for me is the methodology part, which is somewhat not innovative and not clear. It seems that the authors applied a foundation model (pre-trained model) Enformer (it seems to be developed by other group) and fine-tuned it to the novel cell types for epigenetics mark prediction.

2.1 On reflection, we agree with the reviewer that the main advancement of our work was not very clear. We used Enformer as a pretrained model and fine-tuned it to be used on the new task of accurately predicting epigenetic signals in previously unseen cell types. This task is done by using chromatin accessibility data from the cell type/state of interest. Note, that this has not been done successfully previously, for example see our comparison against the current best-in-class approach Epitome (Fig. 3a). Moreover, our fine-tuning included substantial architecture (Additional file 1: Fig. S1) and training methodological developments (methods section Enformer Celltyping architecture). It is innovative in the custom approach to incorporating chromatin accessibility, both a global and local representation, with DNA sequence to make such predictions and results in best-in-class performance at the task (Fig. 3a). Furthermore, we highlight how the differing modelling

choices including the pre-training step resulted in improved performance (Additional file 1: Fig. S3-5), giving insight into how to best approach this style of genomic fine-tuning task. We have updated the manuscript, specifically the results and methods section to better reflect the advancements and utility of Enformer Celltyping.

2.2 Also I can not quite get the point that for a totally novel cell type, if there are not any training data available for this cell type, How the model can be fine-tuned? It seems to be a zero-shot learning scenario and a route fine-tuning strategy is not applicable here. I strongly suggest the authors to reformulate the description of the methodology part to clarify this point.

2.2 Our apologies to the reviewer, we believe that our manuscript was not clear enough previously to quickly deduce this information. We hope that our rewrite in line with points 1.1 and 1.2 will help with understanding of this. Furthermore, to clarify, our approach requires chromatin accessibility (ATAC-seq) data from the cell type of interest, along with DNA to predict the epigenetic signal. So it is useful in the sense that researchers can just perform ATAC-seq on the cell type of interest, or use the multitude of publicly available ATAC-seq datasets for cell types not assayed for other epigenetic types, and get multiple epigenetic signals based on this from Enformer Celltyping and can then also inspect regulatory effects of DNA changes like SNPs. To note, most genomic deep learning models which can't predict in previously unseen cell types have been trained on large repositories like ENCODE's. The issue is that these repositories do not contain assays for all disease-relevant cell types – they mostly have cell lines or bulk tissues assayed. Researchers hoping to study the epigenome and the effect of genetic variants on cell or sub-cell types' not captured in repositories like ENCODE, can not use current genomic deep learning models like Enformer or have to use the closest available cell type from current model's outputs as a proxy. Here, we take a step toward addressing this problem by creating Enformer Celltyping, a model which can predict epigenetic signals in previously unseen cell types, using a large receptive field to capture genomic regulatory information. We show the utility of this by predicting in isolated brain cell types that were not present in ENCODE and so models like Enformer could not predict in (Fig. 5). So, in this sense, Enformer Celltyping does not perform zero-shot learning and requires fine-tuning and additional architecture, and model inputs (chromatin accessibility data) to enable predictions in new cell types. We have made efforts to make this point more clear throughout the manuscript, for example see paragraph starting on line 106 in the introduction and the first paragraph of the results starting on line 537.

3. The limitation of the work, as the authors discussed, is clear that it can not accurately predict genetic variant effect.

3. We agree, overall, Enformer Celltyping's performance in predicting the effect of genetic variants predominately did not replicate the QTL results using our framework with the SLDP measure of concordance (Fig 6a). We hypothesised that this is caused by genomic deep learning models, including Enformer Celltyping, inherently struggling to accurately predict the effect of genetic variants based on their current training paradigm. However, we did not test this on any other model to see if this is likely the case or was the result caused by some other issue specific to Enformer Celltyping. So, to address this, we also ran our genetic variant effect prediction benchmarking framework on Enformer (Fig. 6b). However, Enformer performed similarly poorly to Enformer Celltyping, only correctly identifying one

of six matched association. This agrees with our hypothesis that the real limitation is the lack of genetic variation seen in the current approach to genomic deep learning model's training and is not an isolated issue for Enformer Celltyping. It is also worth noting that Sasse et al. comprehensively showed how Enformer underperforms at predicting the effect of genetic variants, even predicting the incorrect direction of effect in up to 40% of tested cases. The authors came to the same conclusion as our result, hypothesising that this was caused by Enformer being trained on points across the reference genome and then needing to make out-of-sample predictions for individuals' personal genetic variants at any locus. Therefore, our results agree with Sasse et al., that future work hoping to use machine learning models to predict the effect of genetic variants should concentrate on training models on data which accounts for genetic variants. Thus, our result is an important one for the field which will hopefully result in a paradigm shift in future genomic deep learning model training. See discussion section from line 983 for our updated discussion of this point. Furthermore, we have added new analysis showing how, despite genomic deep learning models' limitations in the prediction of variant effects, Enformer Celltyping predictions capture genetic enrichment of complex traits (Fig. 8). This work highlights how our model can be used along with the breadth of publicly available ATAC-Seq data to derive histone mark signals which harbour greater enrichment for genetic variants associated with complex traits and diseases than chromatin accessibility alone, demonstrating its utility for the study of complex traits and disease. We hope this convinces the reviewer of the utility of the work and our attempt to better contextualise the current limitations in the field.

Sasse, A. et al. How far are we from personalized gene expression prediction using sequence-to-expression deep neural networks? 2023.03.16.532969 Preprint at <https://doi.org/10.1101/2023.03.16.532969> (2023).

Reviewer #2 (Remarks to the Author):

In this paper, the authors present "Enformer Celltyping," a deep learning model designed to assess the impact of genetic variants on the epigenome. This model stands out for its ability to predict epigenetic signals irrespective of cell type and considers distal DNA interactions up to 100,000 base pairs away, an improvement over traditional methods. Enformer Celltyping demonstrates more performance compared to existing approaches, showing some robust results across various cell types and biological regions, even those not included in the initial training. The model also includes an application in regulatory quantitative trait loci mapping, highlighting its broad utility in genomic studies. Using a custom transfer learning approach derived from a pre-trained Enformer model, the system efficiently addresses the distal effects of genetic codes on regulatory signals, saving significant memory and training time. Additionally, Enformer Celltyping incorporates both genome-wide and local representations of cell types based on chromatin accessibility, adopting techniques from NLP to those used in the Avocado. This integration makes the model applicable to other cell types with available accessibility data. Validation of Enformer Celltyping has been carried out against leading methods in various biological settings. Overall, this paper represents a contribution to the field of genomics, addressing the challenge of profiling epigenetic effects across diverse cell types. The model's ability to predict epigenetic signals independently of cell type and to consider distal DNA interactions marks an advancement.

We thank the reviewer for their succinct overview of our model and paper.

1. However, this manuscript could be strengthened by addressing some concerns. Firstly, to further substantiate the model's robustness and broad applicability, it would be beneficial to observe its performance on additional, well-characterized cell lines within the ENCODE project, such as the MCF-7 breast cancer line. Testing the model on MCF-7 would not only confirm its efficacy across varied cellular environments but also underscore its utility in cancer research—a field profoundly influenced by epigenetic modifications.

We would like to thank the reviewer for this suggestion and agree that more focus on cancer cell lines would be beneficial for the work; highlighting the model's robustness and broader applicability.

To do this, we wanted to consider cancer cell lines which the model didn't see during the training period - MCF-7 (EpiMap ID: BSS01226) and some other cancer cell lines like GM12878, A549, HEPG2 and HeLa-S3 were included in the training/validation data so we could not use these to measure genome-wide predictive performance. However, not all cell lines from ENCODE were, for example K562 (EpiMap ID: BSS00762) and HEK-293 (EpiMap ID: BSS00526) weren't, so we tested on these. Enformer Celltyping showed no deterioration in its predictive quality compared to the immune cells lines, highlighting it's efficacy at predicting histone mark signals across varied cellular environments (see Additional file 1: Fig. S6a/b and line 744 in the manuscript).

2. Moreover, given the model's demonstrated success in predicting enhancer activities, extending its application to super-enhancers in the same MCF-7 cell line could be significant.

To summarise, super-enhancers are enhancer regions with a multitude of TF binding sites, often occupied by master transcription factors/coactivators like MED1 and BRD4 which are fundamental to cell type-identity. We agree that this would be an interesting avenue of fine-tuning for the model given its strong performance at histone mark signal prediction in enhancer regions (see updated Fig. 4a and Additional file 1: Fig. S6b).

We trained a random forest classifier to distinguish super-enhancers from normal enhancers in these multiple cancer cell lines, using four-fold cross-validation, holding out all data from one cell line in each fold. The super-enhancer regions were obtained from the dbSUPER database (<https://asntech.org/dbsuper/>) for the cell lines MCF-7, K562, HEPG2 and GM12878. While the identification of all enhancer regions was from their ENCODE H3K27ac bed files. We down-sampled the number of enhancer regions (negative samples) to match the number of super-enhancers (positive samples) in the cell line and visualised performance on the held-out test cell line with receiver operating characteristics (ROC) curves (see Methods section 'Super-enhancer prediction in cancer cell lines' for more details).

This analysis highlighted that these super-enhancers are highly cell type-specific (Additional file 1: Fig. S6c), so a method to predict cell type-specific super-enhancers would be useful, highlighting the benefit of Enformer Celltyping's ability to predict in previously unseen cell types. Enformer Celltyping's embeddings showed strong performance on these held-out cell lines (mean area under ROC of 0.85) (Additional file 1: Fig. S6d). We thank the reviewer for instigating this really interesting analysis and findings.

Reviewer #3 (Remarks to the Author):

a. In this manuscript, Murphy et al presented Enformer Celltyping, a model that predicts histone marks by incorporating a recent sequence-based model, Enformer, and cell type-specific ATAC-seq features. Additionally, the authors used *in silico* mutagenesis to identify histone QTLs. The model architecture design presents a new way to combine pre-trained sequence-based model with cell-type-specific finetuning steps. Although the model improved the cell type feature beyond the Enformer model, and outperforms a baseline Epitome, the model still seems to lack sufficient accuracy for predicting histone modifications, and lacks performance to recapitulate hQTLs from experiments.

a. We would firstly like to thank the reviewer for their in-depth review of our work. We also thank them for this overview and agree that the model architecture presents a new way to combine pre-trained sequence-based models with cell type-specific finetuning steps. However, we respectfully disagree that it lacks sufficient accuracy for predicting histone modifications: Enformer Celltyping out-performed the current best-in-class model at the task (Fig. 2a), showed substantially improved performance than an average signal based on Pearson R in functional regions such as distal enhancers (Fig. 4b) and performed consistently well even in cancer cells and cells assayed outside of ENCODE (Fig. 5, Additional file 1: Fig. S6). However, we agree with the reviewer in the sense that this point was not clear in the manuscript and required further analysis which we have now completed (the results from some of which are mentioned above). A lot of these modifications were based on this reviewer's feedback so we would like to thank them for that.

Secondly, we agree that Enformer Celltyping's performance in predicting the effect of genetic variants predominately did not replicate the QTL results using our framework with the SLDP measure of concordance (Fig 6a). We hypothesised that this is caused by genomic deep learning models, including Enformer Celltyping, inherently struggling to accurately predict the effect of genetic variants based on their current training paradigm. However, we did not test this on any other model to see if this is likely the case or was the result caused by some other issue specific to Enformer Celltyping. So, to address this, we also ran our genetic variant effect prediction benchmarking framework on Enformer (Fig. 6b). However, Enformer performed similarly poorly to Enformer Celltyping, only correctly identifying one of six matched association. This agrees with our hypothesis that the real limitation is the lack of genetic variation seen in the current approach to genomic deep learning model's training and is not an isolated issue for Enformer Celltyping. It is also worth noting that Sasse et al. comprehensively showed how Enformer underperforms at predicting the effect of genetic variants, even predicting the incorrect direction of effect in up to 40% of tested cases. The authors came to the same conclusion as our result, hypothesising that this was caused by Enformer being trained on points across the reference genome and then needing to make out-of-sample predictions for individuals' personal genetic variants at any locus. Therefore, our results agree with Sasse et al., that future work hoping to use machine learning models to predict the effect of genetic variants should concentrate on training models on data which accounts for genetic variants. Thus, our result is an important one for the field which will hopefully result in a paradigm shift in future genomic deep learning model training. See discussion section from line 983 for our updated discussion of this point. Furthermore, we have added new analysis showing how, despite genomic deep learning models' limitations in the prediction of variant effects, Enformer Celltyping predictions capture genetic enrichment of complex traits (Fig. 8). This work highlights how our model can be used along with the breadth of publicly available ATAC-Seq data to derive histone

mark signals which harbour greater enrichment for genetic variants associated with complex traits and diseases than chromatin accessibility alone, demonstrating its utility for the study of complex traits and disease. We hope this convinces the reviewer of the utility of the work and our attempt to better contextualise the current limitations in the field.

Sasse, A. et al. How far are we from personalized gene expression prediction using sequence-to-expression deep neural networks? 2023.03.16.532969 Preprint at <https://doi.org/10.1101/2023.03.16.532969> (2023).

Main comments:

1.1 Enformer has histone ChIP-seq as training data. Examining the representative tracks in Figure 1, it seems Enformer Celltyping prediction have both false positives and false negatives.

1.1 We believe the reviewer is referring to the differences in the shape of the predicted histone mark signal and the experimental signal in Fig. 1d. We agree that Enformer Celltyping does not predict the histone mark tracks perfectly, however, it does mostly capture the correct 'shape' of the experimental data, highlighting how the model picks up the regulatory information from the inputted data to predict these protein binding events (albeit not to the same level on every occasion). We believe it shows very strong performance at this. Note that perfectly capturing the experimental signal is often not possible. One reason for this is experimental variability - even two ChIP-Seq replicate tracks of the same assay and cell type will not perfectly resemble one another. Thus, capturing the experimental signal perfectly is not trivial and has not really been done by any genomic deep learning model in the field. For example, see Fig. 1 of the Enformer paper where the model similarly missed the size and in certain places the peak completely in the experimental signal:

1.2 Does the finetuned model improve over the original Enformer? If so, how much does the performance improvements of Enformer Celltyping over Epitome (and original Enformer) come from the original Enformer component?

1.2 Unfortunately, Enformer Celltyping can't be compared against Enformer as Enformer is only trained on held out chromosomes not held out cell types so doesn't have to infer differing cell type specific effects. For more information on this, see the review of these cross-chromosome (e.g. Enformer) and cross-cell type (e.g. Enformer Celltyping) prediction approaches by Schreiber et al., 2020. We note this may not have been clear from our manuscript so have updated throughout to make this point more obvious. However, in short, our approach requires chromatin accessibility (ATAC-seq) data from the cell type of interest, along with DNA to predict the epigenetic signal. So it is useful in the sense that researchers can just perform ATAC-seq on the cell type of interest, or use some of the

multitude of publicly available ATAC-Seq data in cell types which isn't available for other epigenetic marks, and get multiple epigenetic signals based on this from Enformer Celltyping and can then also inspect regulatory effects of DNA changes like SNPs. To note, most genomic deep learning models can't predict in previously unseen cell types and have been trained on large repositories like ENCODE's. The issue is that these repositories do not contain assays for all disease-relevant cell types – they mostly have cell lines or bulk tissues assayed. Researchers hoping to study the epigenome and the effect of genetic variants on cell or sub-cell types' not captured in repositories like ENCODE, can not use current genomic deep learning models like Enformer or have to use the closest available cell type from current model's outputs as a proxy. Here, we take a step toward addressing this problem by creating Enformer Celltyping, a model which can predict epigenetic signals in previously unseen cell types, using a large receptive field to capture genomic regulatory information. We show the utility of this by predicting in isolated brain cell types that were not present in ENCODE and so models like Enformer could not predict in (Fig. 5). We have made efforts to make this point more clear throughout the manuscript, for example see paragraph starting on line 106 in the introduction and the first paragraph of the results starting line 537. Hopefully this explains why we can not compare performance on the histone mark predictions against Enformer. We do, however, compare the performance on genetic variant effect predictions against Enformer using our genetic variant effect prediction framework which we have newly added (Fig. 6a-b). This led to a key finding for the field that we discussed in point a above: the current genomic deep learning model training approach (going across the genome) does not lead to models that can accurately predict the effect of genetic variants.

On “an understanding of how much of the model performance comes from the Enformer component”, we did look into this when considering Enformer Celltyping's genome-wide predictive performance (Fig. 3b). Here, one of the approaches we benchmark against is EC Avg which is the output of the Enformer architecture used in Enformer Celltyping when it is fine-tuned, without the chromatin accessibility signal, to predict the histone mark levels. This comes from the pre-training step we used for Enformer Celltyping (see the Methods section Enformer Celltyping architecture for more information on this). However, note that this is predicting an average signal across all training cells so is not trying to predict in a cell type-specific manner (since it only has DNA sequence as input). Enformer Celltyping's cell type-specific predictions outperform that of this EC Avg in the vast majority of cell types and histone marks and also, the variance in the error of the predictions genome-wide is far smaller, indicating how Enformer Celltyping picks up on cell type-specific signals even at genome-wide predictive performance comparisons (Fig. 3b). This highlights the benefit of the supplementary architecture Enformer Celltyping has at adding additional regulatory information across cell types when making these predictions in new cell types. I don't think this point was very clear in the manuscript so I have updated it to reflect this. For example, see from line 626 in the results section and for a description of the pre-training step see the methods section “Enformer Celltyping architecture”. I hope both of these now clearly highlight these points.

Schreiber, J., Singh, R., Bilmes, J. et al. A pitfall for machine learning methods aiming to predict across cell types. *Genome Biol* 21, 282 (2020). <https://doi.org/10.1186/s13059-020-02177-y>

2. The main evaluation metric is MSE. However, MSE may not be a good metric because it is affected by data scale. Correlation metric like Pearson/Spearman correlation at bin level will be more informative. Moreover, general performance evaluation across a few independent samples, especially focusing on the cell-type-specific genomic regions, will better present the model performance. One way to evaluate is to merge all peak loci from both experimental data and predicted data across multiple cell types, and evaluate the performance in these differential group.

2. We agree with the reviewer that Pearson/Spearman correlation at the bin level along with focusing on the cell-type-specific genomic regions would be beneficial and give additional information on our model's performance. As such, we have measured the model's Pearson R against the true experimental values for functional genomic regions (promoters, proximal and distal enhancers from SCREEN) on histone marks known to be active in these regions – promoters; H3K27ac, H3K4me3, proximal enhancers; H3K27ac and H3K4me1, and distal enhancers; H3K27ac and H3K4me1 (Fig. 4). We have also updated the text to reflect this decision (see from line 662. Moreover, we changed to use Pearson R as part of the analysis of cell types derived outside of Encode; the Nott et al. brain cell types (Fig. 5) and also for the analysis into cancer cell predictive performance (Additional file 1: Fig. S6). We believe these new results shows a better representation of Enformer Celltyping's capabilities, outperforming the average signal and showing strong Pearson R correlations, especially in distal enhancer regions which would be particularly cell type-specific which shows a key benefit of the model's use. We thank the reviewers for this suggestion.

To note previously, when considering functional regions from SCREEN, we binned the genome by our model's predictive field and then added the MSE for the bin to the different genomic regions if they overlapped the bin. However, these genomic regions are very small in comparison to the model's predictive field – for example, the median size of distal enhancer regions from SCREEN was 294 base-pairs versus our model's predictive field of 114,688 base-pairs, which resulted in a lot of off-target regions being included. Moreover, previously, we also included predictions for histone marks which had no association with the functional region such as H3K4me1, a mark of active enhancers, in promoter regions. We believe the updated analysis gives a more true measure of Enformer Celltyping's performance in the functional regions.

3. In Figure 2, was the microglia peak validated in experiments? Plotting experimental tracks across the samples and compare them with predictions will help better evaluate the model performance.

3. Yes, the microglia peak discussed in Fig.2 is present in the experimental track. Originally, the idea of Fig. 2 was to highlight that the model makes distinct, cell type-specific predictions (not just predicting the same signal for every inputted cell type). Figure 1, 3, 4 and 5 all highlight the model's performance in many marks genome-wide and across differing functional regions which captures the experimental signal so we didn't believe this is necessary here. However, we agree that it seems odd now the way Fig. 2 is presented so we have added the experimental tracks for both the immune cells and microglia to show the model's prediction is correct. We hope this answers the reviewer's concerns on this point and thank them for this suggestion.

5. In Figure 4 the No Peak region, it seems neutrophil and T-cell-specific predictions have higher MSE than average. Why is that? Does it indicate bad prediction quality in the background, meaning more false positives? The authors can show a confusion matrix heatmap for every cell type so that people can better understand the performance difference between cell types.

5. We thank the reviewer for bringing this to our attention for Fig. 4a. However, we don't believe a confusion matrix would not be suitable here since we are predicting a regression task; the $-\log_{10}$ p-value. Although the 'no peak' category was defined by a $-\log_{10}$ p-value cut-off thus creating a 0/1 peak/no peak split, it still makes more sense to consider a measure like MSE or Pearson R which tells not only if a peak was found or not found but also how close to the real value the model got. However, we agree that in the 'no peak' category, there is better performance on monocytes than the other two immune cells. However, the opposite is true in the other categories. Like the reviewers state, this appears to be a systematic difference which appears to be because the model is making a trading-off with worse predictions in no peaks but better predictions in the peaks/SCREEN functional regions. For example, see H3K36me3; No peak/peak or functional region for neutrophil and T-cell vs average and monocyte. This could be that the model is making more conserved predictions for monocytes which results in better performance in the no peak regions but worse in functional regions.

6. When performing mutagenesis, did the authors modify only the DNA sequence without changing ATAC-seq signal? That would be a problem because the ATAC signal was likely to change as a result of the sequence change. In that case, the model could output invalid predictions.

6. Yes, we agree this is a limitation of models that use alternative genomic data as input when predicting the effect of genetic variants which we discuss as a key limitation of the work. We have added to the discussion section (line 1022) to further emphasise this point. However, we don't believe this is the reason Enformer Celltyping does not capture the genetic variant effect predictions based on the QTL data since we also tested Enformer which performed similarly poorly (Fig 6b) – see the second paragraph of our response to a above for more information on this result but in short, we agree with the view of others in the field that current genomic deep learning models inherently struggle to accurately predict the effect of genetic variants based on their current training paradigm. This is a key result to be seen by those working in the field.

7. When masking global accessibility signal to evaluate model's reliance on the global accessibility, did the author train a separate model (without global accessibility) or use the original model but masking it during prediction? If using the original model, it can produce invalid predictions as well since the masking process was unseen during training.

7. For this analysis, we did not train a separate model and instead just used the trained model, predicting while masking the global chromatin accessibility. This analysis was done to inform where the global signal was important so we first predicted genome-wide as normal and identified all peaks for the histone marks then predicted genome-wide without the global signal and checked, for the same peaks, which decreased the most, i.e. which peaks were most reliant on the global signal. However, we agree that this approach is not ideal so have redone the analysis. This time, for each peak, the influence of the global signal was approximated by calculating the partial derivatives of the model with respect to the

input, i.e. the gradient on the input (see Figure 7 and methods section “Global chromatin accessibility signal – cell type-specific motif enrichment”). We thank the reviewer for this suggestion.

8. In Figure 7, there are many missing hits, and the predicted results does not seem to be interpretable. For example, in monocyte group, the highest hit is unexpectedly categorized as astrocyte. The other screening results also seemed very mixed. How to understand these results and do they make sense?

8. We agree that there were many significant results in Figure 7 which had unclear meaning – we believe this was, at least in part, due to our previous approach of simply zeroing out the global signal. Now that we swapped to using a more rigorous approach of calculating the partial derivatives with respect to the global chromatin accessibility input, we can see there are far fewer significant results (Figure 7). Moreover, the results, for the most part, seem to make better sense – for example, the analysis of monocytes showed significant enrichment in Myeloid cells, heart which contains epithelial cells in the epicardium had the largest enrichment in corneal epithelial cells and neuron shows strong enrichment in the neuron 3 cell type. However, we admit that these results, although better, are still not perfect with a limited number of expected significant enrichments, which could be due to confounders such as the limited number of transcription factors with known motifs. Moreover, mapping these back to human based on orthologs and the fact that some motifs will span whole transcription factor families may affect the quality of the results. We discuss these limitations in the discussion (paragraph starting 950).

Technical comments:

1. Does predicting p-value make sense? The distribution of p value is weird and can interfere with training process. One alternative is \log_2fc .

1. We believe the reviewer has highlighted a very reasonable question which we answer in detail below but we also feel is relevant to the general reader so have added it to a supplementary note with the paper. We thank the reviewer for raising this.

We agree that using the fold change output from ENCODE’s MACS2 processing pipeline (\log_2fc) has its benefits and has been used by other models (such as Avsec et al., 2021 and Kelley DR, 2020). Equivalently many models, especially those imputing epigenetic signals in previously unseen cell types, use the $-\log_{10}$ p-value signal (Schreiber et al., 2020, Ernst and Kellis, 2015 and Durham et al., 2018). Moreover, when considering the differences between the two, the $-\log_{10}$ p-value was found to have better signal-to-noise properties and so, was thought to be more suitable. For this reason, ENCODE chose $-\log_{10}$ p-value as the primary signal tracks for analyses in (Roadmap Epigenomics Consortium et al., 2015). This is discussed in more detail by (Ernst and Kellis, 2015):

“We selected the $-\log_{10}$ P value signal tracks rather than the fold-change tracks for histone marks and DNase as they were designated the primary signal tracks for analyses in (Roadmap Epigenomics Consortium et al., 2015) on the basis of having better signal-to-noise properties. In particular, both sets of tracks were generated based on downsampling highly sequenced datasets to the same sequencing depth, thus in the $-\log_{10}$ P value track, no dataset had a disproportionately high signal simply due to being sequenced very deeply, whereas on the other hand under-sequenced datasets were included and in some cases had

locations with high fold-change signals that were the result of noise and did not have values as relatively high on the $-\log_{10}$ P-value track.”

We hope, based on this, it is clear why we chose to model the $-\log_{10}$ p-value with Enformer Celltyping especially since we are considering predictions across cell types. Moreover, $-\log_{10}$ p-values are more interpretable which is useful for example, thresholding for peaks such as what ChromHMM does (Ernst and Kellis, 2015). This can't be easily done with logfc values. We agree that this was not clear from the text so we have added additional commentary about this to the discussion (line 955) and introduction (line 153).

In our work here, we also went a step further in addressing the “disproportionately high signals” in some genomic locations that were less common but still present in the $-\log_{10}$ P-value. We processed the p-value signal using an arc-sinh transformation to help with differences in sequencing depth as others have done (Schreiber et al., 2020 and Durham et al., 2018). The arc-sinh transformation ‘tows in’ extreme values so these do not affect the model training (see line 162 of manuscript), see Figure below:

$$\sinh^{-1} x = \ln \left(x + \sqrt{1 + x^2} \right)$$

Also please see Fig. s11 in the supplementary of the manuscript (Additional file 1: Fig. S11) where we show the strong effect the transformation has: The arcsinh-transformed data shows clear separation between assay types (right), something which is not present with the raw data (left), highlighting arcsinh-transform's benefit.

- Avsec, Ž., Agarwal, V., Visentin, D. et al. Effective gene expression prediction from sequence by integrating long-range interactions. *Nat Methods* 18, 1196–1203 (2021). <https://doi.org/10.1038/s41592-021-01252-x>
- Kelley DR (2020) Cross-species regulatory sequence activity prediction. *PLoS Comput Biol* 16(7): e1008050. <https://doi.org/10.1371/journal.pcbi.1008050>
- Schreiber, J., Durham, T., Bilmes, J. et al. Avocado: a multi-scale deep tensor factorization method learns a latent representation of the human epigenome. *Genome Biol* 21, 81 (2020). <https://doi.org/10.1186/s13059-020-01977-6>
- Ernst, J., Kellis, M. Large-scale imputation of epigenomic datasets for systematic annotation of diverse human tissues. *Nat Biotechnol* 33, 364–376 (2015). <https://doi.org/10.1038/nbt.3157>
- Durham, T.J., Libbrecht, M.W., Howbert, J.J. et al. PREDICTD PaRallel Epigenomics Data Imputation with Cloud-based Tensor Decomposition. *Nat Commun* 9, 1402 (2018). <https://doi.org/10.1038/s41467-018-03635-9>

- Roadmap Epigenomics Consortium., Kundaje, A., Meuleman, W. et al. Integrative analysis of 111 reference human epigenomes. Nature 518, 317–330 (2015). <https://doi.org/10.1038/nature14248>
- Ernst, J., Kellis, M. Chromatin-state discovery and genome annotation with ChromHMM. Nat Protoc 12, 2478–2492 (2017). <https://doi.org/10.1038/nprot.2017.124>

2. How does ATAC-seq data quality affects prediction? How does subsampling the bam files for varied depth affects prediction?

2. We agree that this is an important analysis to conduct to give users an idea of the expected performance when chromatin accessibility data for their cell type of interest is only available at a lower sequencing depth than what we trained Enformer Celltyping on. We investigated the chromatin accessibility (ATAC-seq) data sequencing depth, an input for Enformer Celltyping, by repeating a down-sampling analysis 5 times with differing seeds on the isolated oligodendrocytes from Nott et al. 2019 (Fig. 5c). We measured performance based on Pearson R in distal enhancer regions, from SCREEN, as they are known to be cell type-specific. Even at a sequencing depth of 25 million, 50% of what the model was trained to predict with, Enformer Celltyping outperforms the average signal in distal enhancer regions, highlighting the model's applicability under data quality variations and giving guidance on expected performance with differing sequencing depths in experimental data for users. We thank the reviewer for this suggestion.

3. In Sup Fig 1a when flatten the cell type embeddings, how was it performed? Was the position information affected in this process?

3. The embedding layers convert the chromatin accessibility information from 1D to 2D shape where the size of the second dimension is based on the number chosen by the user. The embedded vector representation is then updated through backpropagation to better reflect the inputted signal. Flattening after the embedding layers will not affect the positional information of the chromatin accessibility information. See the example below to explain and motivate flattening after embedding and how positional information was not affected:

Say we have two sentences as our entire dictionary:

Sentence 1 I play football
Sentence 2 I play basketball

and so we assign integers to the words as follows:

Sentence to sequence
Sequence 1 [1, 2, 3]
Sequence 2 [1, 2, 4]

Then we take the input (the integer sequences) and add them to a Embedding layer which assigns random numbers in 2 dimensions and using backpropagation to update these values:

Embedding matrix:

Word	Index	Vector
I	1	[0.4, 0.2]
play	2	[0.7, 0.3]
football	3	[0.2, 0.1]
basketball	4	[0.2, 0.8]

So the embedding of the sentences looks like:

Embedding:

Sequence 1 [[0.4, 0.2], [0.7, 0.3], [0.2, 0.1]]

Sequence 2 [[0.4, 0.2], [0.7, 0.3], [0.2, 0.8]]

Next the model flattens the sequences with the embeddings in order to make it 1 dimension again:

Flatten to 1 dimension:

Sequence 1 [0.4, 0.2, 0.7, 0.3, 0.2, 0.1]

Sequence 2 [0.4, 0.2, 0.7, 0.3, 0.2, 0.8]

I hope this is clear how the words above's order remains the same after embedding and flattening which similarly, the order of the chromatin accessibility data is similarly preserved. This same flattening approach has been used in other genomic deep learning models like Avocado (<https://genomebiology.biomedcentral.com/articles/10.1186/s13059-020-01977-6>, model architecture: <https://github.com/jmschrei/avocado/blob/master/avocado/model.py>) and is also used commonly in NLP without affecting positional information such as Google research's BUSTLE model (<https://arxiv.org/abs/2007.14381>, model architecture (https://github.com/google-research/google-research/blob/6ca244220d175ace3bcd6e2167b730fe0f95a2ee/bustle/bustle_python/train_model.py)). We have updated the text of the manuscript to reflect an expanded explanation of this at line 311.

4. Is the p-val calculation in Nott et al. the same as Encode pipeline? Otherwise it would be difficult to justify model performance across databases.

4. Thank you for pointing this out, yes, we did use the same ENCODE pipeline to generate the p-value bigwigs so they are the same as the training data. This is an important step to ensure consistent processing of the data. I have added to the results section to explain this more clearly (line 698) and in the methods (line 169).

Minor issue: 884: "ffect" Typo.

This has been corrected, thank you!

Reviewer #4 (Remarks to the Author):

I co-reviewed this manuscript with one of the reviewers who provided the listed reports.

This is part of the Nature Communications initiative to facilitate training in peer review and to provide appropriate recognition for Early Career Researchers who co-review manuscripts. We thank this reviewer for their efforts co-reviewing to facilitate training in peer review for ECRs.

Reviewers' comments:

Reviewer #3 (Remarks to the Author):

- The reviewer appreciate the authors' efforts in responding the comments and concerns in depth, and is fully aware of the importance of cell-type-specific predictions. However, the reviewer is worried that core issues/concerns, particularly regarding model accuracy, was not clearly addressed, which would not make the model truly achieve the expected function for predicting cell type-specific epigenetic signals.

- We thank the reviewer for this and their time and insight into our work. We would also like to thank the reviewer for highlighting these issues as, we are sure, some readers would be of the same opinion, thus warranting further insight and discussion. We believe our comments below and our noted updates to our manuscript will cover these concerns.

1.1 Concerns on the model's performance/accuracy. It does not seem the authors addressed the concerns regarding the inaccuracy of the model prediction. The authors agreed that the Enformer Celltyping's performance is similar to the average signal of the training dataset, noticing the very minimal difference between "Enformer Celltyping" and "Avg" groups in Figure 3b.

1.1 We believe there may be some misinterpretation of the results from our first review update. Yes, we noted that there was little performance difference between "Enformer Celltyping" and "Avg" groups in the test set, the blood immune cell types, in Figure 3b. However, this was the genome-wide measure of performance which an average signal would be expected to perform fairly strongly in. To analyse this performance comparison further, and on the back of this reviewer's very useful suggestions in the last review cycle, we considered the performance of our model against the average in functional regions, using the Pearson R performance measurement - **Fig. 4c, Fig. 5b** and **Supplementary Fig. S7b**. Here, we focused on histone marks with known functional roles in these regions; H3K27ac at active enhancers and promoters, H3K4me3 at active promoters and H3K4me1 at active enhancers, and derived the biological regions from Search Candidate cis-Regulatory Elements by ENCODE (SCREEN).

In these comparisons, we can see Enformer Celltyping far out-performs the average model in cell type-specific regulatory regions. This is particularly notable for distal enhancer regions in the blood immune cell types (**Fig. 4c, panel 3**) where it was better in 96% of the cell type-chromosome comparisons. Moreover, in the cell types derived from Nott et al., i.e. outside of ENCODE (**Fig. 5b**), for three of the four cell types it was better in at least 95% of comparisons. And finally, in proximal and distal enhancer regions for cancer cell lines (**Supplementary Fig. S7b**), Enformer Celltyping was better in 100% and 91% of comparisons respectively. We acknowledge that these comparisons were not clearly explained in our text and so have emphasised the point in the manuscript (see line 675, line 730 and line 766). For example on line 675:

"This is particularly notable for distal enhancer regions for the blood immune cell types where Enformer Celltyping outperformed the average in 96% of the cell type-chromosome comparisons (Fig. 4c, panel 3)."

These analyses highlight Enformer Celltyping's superior predictive ability in highly cell type-specific genomic loci, demonstrating its utility for the field.

To further emphasize the point of Enformer Celltyping's superior performance in the distal enhancer regions for the immune cell types and to aid in comprehension, we have also plotted the experimental, predicted and average track in some of these regions, highlighting loci where the average fails to capture or predicts a peak which isn't present in the experimental track whereas Enformer Celltyping does match the experimental track (**Supplementary Fig. S6**). For the interest of the reviewer and readers, we have also made available full genome browser track visualisations for T-Cells H3K27ac, including the experimental track, Enformer Celltyping and average predictions: https://genome-euro.ucsc.edu/s/almurphy/EC_hg19. We have also added this to the data availability section of the manuscript.

From this and our other analyses in **Fig. 4**, **Fig. 5** and **Supplementary Fig. S7**, it is clear Enformer Celltyping does not simply predict the average and outperforms it, managing to capture cell type-specific signals for previously unseen cell types. We hope this is now clear from our discussion here but also in our manuscript.

1.2 Then the authors claimed that “Enformer Celltyping's cell type-specific predictions (Enformer Celltyping) outperform that of the average predictions from pre-training (EC Avg) in the vast majority of cell types and histone marks”, so that to support that “Enformer Celltyping picks up on cell type-specific signals”. It is misleading to claim in this way, as the data only has a minimal difference, similar to the evidence between “Enformer Celltyping” and “Avg” groups mentioned above.

1.2 We agree that this point was not backed-up by our analysis and we thank the reviewer for pointing this out. Firstly, we have updated the text to avoid any possible misleading results and instead, have explained that the difference seen was minimal (see from line 684). Moreover, given that this point was not backed-up, we have added a comparison of our model to the average predictions from pre-training (EC avg) in the functional regions of promoters, proximal and distal enhancers for instigated histone marks (following the same analysis as conducted in the comparison to the training set average in **Fig. 4c**). This has been added to **Fig. 4b** of the manuscript and shows the clear, superior performance of our model against the pretraining step which we hope the reviewer will agree, backs-up our claim that Enformer Celltyping picks up on cell type-specific signals during the full training step.

1.3 It is important to report the confusing metrics and simultaneously to examine the actual performance when compared to the experimental data.

1.3 We believe the reviewer here meant a confusion matrix. As we noted in the previous review round, a confusion matrix would not be suitable here since this is not a classification (0 or 1) but a regression task; predicting the $-\log_{10}$ p-value at differing genomic loci. Thus, it makes more sense to consider a measure like MSE or Pearson R which tells us not only if a peak was found or not but also how close to the experimental value (i.e. number of mapped reads account for the background signal) the model got (MSE) and does the predicted shape match the actual shape (Pearson R). At this point, we have done extensive work to look into Enformer Celltyping's performance compared to the experimental data for 9 different cell types across blood immune cells (**Fig. 3** and **Fig. 4**), brain cells (**Fig. 5**) and cancer cells (**Supplementary Fig. S7b**). We hope the reviewer will acknowledge this effort in place of their request for a confusion matrix.

However, upon reading this comment, we went back reread this reviewer’s original request on this in the first round of reviews which stated: “The authors can show a confusion matrix heatmap for every cell type so that people can better understand the performance difference between cell types.” We would like to apologise as we previously missed the point on comparing model performance across cell types which we agree would be very useful. As such, we have added a heatmap to compare Enformer Celltyping’s performance in the functionally relevant regions across all 9 cell types tested to give insight into this (**Supplementary Fig. S8**). In this figure, we can see the weighted average of the three SCREEN regions for the histone marks for each cell type (last row) and for each SCREEN and histone mark combination (last column). Using these, it is clear that on average, Astrocytes had the worst performance which is consistent with the results in **Fig. 5b** and is likely attributable to the low sequencing depth of the associated ATAC-Seq data. Importantly however, average performance for cell types was approximately a Pearson R of 0.65, even on cells from outside of ENCODE (Microglia and Oligodendrocytes), highlighting Enformer Celltyping’s consistent performance across cell types. We have added this to the manuscript (see line 781) and hope this answers the reviewer’s request for a heatmap comparing performance across cell types.

On the other hand, if the reviewer meant that using multiple performance measures is confusing and should be explained, we agree this could be the case for readers. As such, we have given more insight into why we considered Pearson R and MSE in the results section, line 665. We hope this answers the reviewer’s concerns on this matter.

1.4 The reviewer appreciate the authors adding experimental data tracks during the revision in Figure 2b. Examining the data at the AIF1 locus (in Figure 1d and 2b together), it is clear that the model predicts the best in constitutive ‘active’ regions – regions where there are minimal cell-type-specific signals. However, the Enformer Celltyping model did not correctly capture the expected signals at AIF1 promoter or upstream enhancer regions – false positives and false negatives from the current model prediction are very evident in these regions, and to some extent seems randomly assigned.

1.4 We agree that our model’s predictions aren’t perfect in cell type-specific regions however, as mentioned in the previous review notes, perfectly capturing the experimental signal is often not possible for any genomic deep learning model. One reason for this is experimental variability - even two ChIP-Seq replicate tracks of the same assay and cell type will not perfectly resemble one another. Thus, capturing the experimental signal perfectly is not trivial and has not been done by any genomic deep learning model in the field. For example, see Fig. 1 of the Enformer paper which we previously discussed but is still relevant - the model similarly missed the size and in certain places the peak completely in the experimental signal:

Moreover, and more importantly, the reviewer's insight here is based on just one locus. It is far more informative to consider performance over many functionally relevant regions. We have conducted such analyses, on the back of this reviewer's helpful suggestions in the previous review round, on all cell type-specific enhancer regions genome-wide, across many cell types and for many marks **Fig. 4b-c**, **Fig. 5b-c** and **Supplementary Fig. S7b**. This extensive analyses all showed Enformer Celltyping's strong performance in cell type-specific regions as opposed to just in constitutive 'active' regions, i.e. non-cell type-specific, as the reviewer states.

Moreover, to further emphasize this point to the reviewer, we have added plots of the experimental, predicted and average track in some of these regions, highlighting where the average fails to capture, or predicts a distal enhancer peak which isn't present, in the experimental track whereas Enformer Celltyping does match the experimental track. (**Supplementary Fig. S6**). We hope **Supplementary Fig. S6** gives a visual aid of how the model's predictions look in specific loci compared to the experimental signal but also highlights our point that over-analysis of performance in a single locus is not informative of the model performance as a whole down to its discrepancy to the reviewer's intuition of the model's performance based on **Fig. 1b** and **Fig. 2d**. And we hope, from all of our analyses across the genome for functional, cell type-specific regions (**Fig. 4b**, **Fig. 5b** and **Supplementary Fig. S7**), it is clear that Enformer Celltyping does not simply predict the average and outperforms it, managing to capture cell type-specific signals for previously unseen cell types.

2. how much does the current model improve over the basal model, the original Enformer? Since the Enformer parallelly predicts tracks over many cell types included in the training. The reviewer's original suggestion of comparison to Enformer can easily be performed by comparing the model performance in the Enformer training cell types (which were also used as training cell types in the current Enformer Celltyping model). In cell types that are both included as training data, it is expected that the current model will perform much better – if the current model truly works – than the original Enformer, especially in the regions where Enformer does not perform well.

2. We note that we have answered why Enformer Celltyping's predictive performance should not be compared against Enformer in the previous round of reviews (see comment 1.2 to this reviewer). However, to summarise, yes we used Enformer as a pretrained model in our architecture but, it does not make sense to compare the predictive performance against Enformer. The main reason these models should not be compared is that they perform different, non-comparable tasks - Enformer predicts in held-out chromosomes and so does not have to extrapolate predictions to previously unseen cell types whereas Enformer Celltyping does on whole, held-out cell types. See 'A pitfall for machine learning methods aiming to predict across cell types' Fig. 1a-b for a visual representation of how these tasks differ and a detailed description of both.

The reviewer correctly states that Enformer is a basal or base model of Enformer Celltyping. We do this using transfer learning, a machine learning technique that allows us to repurpose pre-trained models on new task. However, this does not mean it is suitable to compare against the base model in either this new or the old task. To give an example, many models have used pre-trained image classification models trained on generic datasets such as Modified National Institute of Standards and Technology database (MNIST), which is

pictures of handwritten digits, with fine-tuning to apply them to the medical domain. For example, for brain tumour classification from Magnetic resonance imaging (MRIs), see 'Employing deep learning and transfer learning for accurate brain tumor detection'. Yet, comparing the MNIST, pretrained model to the brain tumour model on either task would not be beneficial. In the same way, we should not compare Enformer to Enformer Celltyping. Nevertheless despite this, and as a thought experiment, let's consider trying to compare the two models. If we were to compare Enformer to Enformer Celltyping, we should not do it based on the training set, as the reviewer suggested, as this would just produce inflated performance for both models, potentially misleading the reader. Moreover, we would be interested in the performance in functionally relevant regions (as we discuss for our other comparisons of Enformer Celltyping). Thus, the only option would be to test on the intersection of Enformer's test set and SCREEN functional regions on cell types that Enformer trained on and that Enformer Celltyping did not train on, i.e. a blind test set for both. Overlapping SCREEN regions with Enformer's test set results in a total of just 152 promoter, proximal enhancer and distal enhancer regions. These are very small regions (on average ~143 base-pairs) and the overlapping base-pair positions were just 21,748 bps total or 169 predicted positions since Enformer and Enformer Celltyping predict in 128 bp blocks. This equates to just 0.0001% of chromosome 1 alone. Given this is so small it isn't a representative test set and no insight should be taken from any results based on this. To give some comparison, Enformer tested on 1,937 test regions of predicted window size 114,688, i.e. 222,150,656 or 222 million base-pairs. Also, we tested on 1,034,718 SCREEN positions i.e. 282,041,971 or 282 million base-pairs - approximately 13,000 times more positions. This issue is furthered if we calculate performance values at a chromosome level for the functional regions, to get multiple values to have a range in performance. Some of the chromosomes would have just single values and so would result in making computation of a correlation impossible.

Furthermore, comparing the models at the same positions would not even be possible with the above approach: The SCREEN locations that overlap Enformer would need to be found using SCREEN hg38. However, one would then need to map these regions to hg19 to use for Enformer Celltyping. There is little to say this mapping is accurate or that these resulting regions in hg19 are actually in the same functional regions in SCREEN.

Finally, Enformer predicts fold change and Enformer Celltyping predicts p-values. So the performance comparison wouldn't even be based on the same y values, just at similar positions for similarly named cells types.

From this explanation, we hope it is clear why Enformer Celltyping can not be compared against Enformer. We have however, compared Enformer Celltyping against the current best-in-class model for predicting histone mark signals in previously unseen cell types (i.e. extrapolating predictions to previously unseen cell types) Epitome in **Fig. 3a** where our model showed notably strong performance. Moreover, we have compared Enformer Celltyping to Enformer for the genetic variant effect prediction task, a key task for genomic deep learning models, where they performed similarly (**Fig.6a-b**). We hope this answers the reviewer's concerns. We have also added this explanation as a supplementary note.

Citations:

Schreiber, J., Singh, R., Bilmes, J. et al. A pitfall for machine learning methods aiming to predict across cell types. *Genome Biol* 21, 282 (2020). <https://doi.org/10.1186/s13059-020-02177-y>

Mathivanan SK, Sonaimuthu S, Murugesan S, Rajadurai H, Shivahare BD, Shah MA. Employing deep learning and transfer learning for accurate brain tumor detection. *Sci Rep*. Mar 27;14(1):7232 (2024). <https://doi.org/10.1038/s41598-024-57970-7>

Other comments:

3. The reviewer does not agree with the claim that ENCODE data was not able to train good machine learning models. Many good ML models have been trained with ENCODE data and achieved great performance. It may require more new data for the many more ML training, but the reviewer believes that the current ENCODE data are overall of good and standardized quality for ML applications.

3. We believe this is in relation to this statement in our discussion section (line 963):

“Our reliance here on imputed data highlights the importance of generating new biological datasets specifically designed for training machine learning models: At present, almost all models in the field use ENCODE, which was not created for this purpose.”

We thank the reviewer for pointing this out as we strongly agree with their opinion – that ENCODE has been central to the recent wave of genomic machine learning models. As such, we have updated this sentence to (line 968):

“Our reliance here on imputed data highlights the importance of generating new biological datasets to supplement those already available which have been paramount for the training genomic deep learning models like ENCODE.”

4. There are still many typos in the manuscript. Some of them from the discussion section: Lines 939, 962, 965,

4. We apologise and thank the authors for noting this. We have updated the typos on line 939, 962 and 965:

- 939: it's to its
- 962: our to or
- 965: 'At present almost' to 'At present, almost'.

Moreover, we carefully re-examined the whole manuscript and found further necessary grammatical changes in the introduction/discussion:

- Use of both ATAC-Seq and ATAC-seq throughout corrected to ATAC-Seq
- Some figure numbers stated as Fig.X instead of Fig. X throughout
- Line 67: 'sizes in the high hundreds, and some reaching' to 'sizes in the high hundreds and some reaching'
- Line 104: 'infer the cell type variability' to 'infer cell type variability'
- Line 122: sub-cell types' to sub-cell types
- Line 620: 'would be expected perform' to 'would be expected to perform'

- Line 942: 'we recreated Enformer, removing' to 'we recreated Enformer; removing'
- Line 1045: sub-cell types' to sub-cell types.

Reviewer #3 (Remarks to the Author):

I thank the authors for the detailed explanation, but with all respect to the authors and their continuous efforts, I still hold my concerns about the model's performance/accuracy from the last round of review. I believe that a new ML model (or any other new technology) should provide significant advancement to the field. However, I do not see this model achieving that level of advancement to predict cell-type-SPECIFIC epigenome features beyond the original Enformer model. To elaborate on it, I attached a file, showing the screenshot of Figure 2b in the manuscript, and in comparison, to only show the cell-type-specific regions together with the provided experimental data (by masking cell-type-shared regions and cell types that do not have accompanied experimental data for validation). The remaining results clearly show that the current model predicts 'cell-type-specific' epigenetic signals in an almost random way. Thus I do not believe the current model would make meaningful contribution to real-world applications beyond the original Enformer model.

We thank the reviewer for this note and their work. However, we would like to remind them that **Fig. 2b** is just one genomic loci. To systematically analyse the model's performance, we benchmarked it against the average in functional regions, using the Pearson R performance measurement - **Fig. 4c**, **Fig. 5b** and **Supplementary Fig. S7b**. Here, we focused on histone marks with known functional roles in these regions; H3K27ac at active enhancers and promoters, H3K4me3 at active promoters and H3K4me1 at active enhancers, and derived the biological regions from Search Candidate cis-Regulatory Elements by ENCODE (SCREEN). In these comparisons, we can see Enformer Celltyping far out-performs the average model in cell type-specific regulatory regions. This is particularly notable for distal enhancer regions in the blood immune cell types (**Fig. 4c**, panel 3) where it was better in 96% of the cell type-chromosome comparisons. Moreover, in the cell types derived from Nott et al., i.e. outside of ENCODE (**Fig. 5b**), for three of the four cell types it was better in at least 95% of comparisons. And finally, in proximal and distal enhancer regions for cancer cell lines (**Supplementary Fig. S7b**), Enformer Celltyping was better in 100% and 91% of comparisons respectively. We hope this shows the model's performance better than just one loci.